# Vocalization categorization behavior explained by a feature-based auditory categorization model

**Manaswini Kar[1,2,3], Marianny Pernia[3†], Kayla Williams[3†], Satyabrata Parida[3], Nathan Alan Schneider[1,2], Madelyn McAndrew[2,3‡], Isha Kumbam[3‡], Srivatsun Sadagopan[1,2,3,4,5]***

[1]Center for Neuroscience at the University of Pittsburgh, Pittsburgh, United States; [2]Center for the Neural Basis of Cognition, Pittsburgh, United States; [3]Department of Neurobiology, University of Pittsburgh, Pittsburgh, United States; [4]Department of Bioengineering, University of Pittsburgh, Pittsburgh, United States; [5]Department of Communication Science and Disorders, University of Pittsburgh, Pittsburgh, United States

**Abstract** Vocal animals produce multiple categories of calls with high between- and within-subject variability, over which listeners must generalize to accomplish call categorization. The behavioral strategies and neural mechanisms that support this ability to generalize are largely unexplored. We previously proposed a theoretical model that accomplished call categorization by detecting features of intermediate complexity that best contrasted each call category from all other categories. We further demonstrated that some neural responses in the primary auditory cortex were consistent with such a model. Here, we asked whether a feature-based model could predict call categorization behavior. We trained both the model and guinea pigs (GPs) on call categorization tasks using natural calls. We then tested categorization by the model and GPs using temporally and spectrally altered calls. Both the model and GPs were surprisingly resilient to temporal manipulations, but sensitive to moderate frequency shifts. Critically, the model predicted about 50% of the variance in GP behavior. By adopting different model training strategies and examining features that contributed to solving specific tasks, we could gain insight into possible strategies used by animals to categorize calls. Our results validate a model that uses the detection of intermediate-complexity contrastive features to accomplish call categorization.

**\*For correspondence:**
vatsun@pitt.edu

†These authors contributed equally to this work
‡These authors also contributed equally to this work

**Competing interest:** The authors declare that no competing interests exist.

## Editor's evaluation

This important study combines behavioral data from guinea pigs and data from a classifier model to make a compelling case for which auditory features are important for classifying vocalisations. This study is likely to be of interest to both computational and experimental neuroscientists, in particular auditory neurophysiologists and cognitive and comparative neuroscientists. A strength of this work is that a model trained on natural calls was able to predict some aspects of responses to temporally and spectrally altered cues.

## Introduction

Communication sounds, such as human speech or animal vocalizations (calls), are typically produced with tremendous subject-to-subject and trial-to-trial variability. These sounds are also typically encountered in highly variable listening conditions—in the presence of noise, reverberations, and competing

sounds. A central function of auditory processing is to extract the underlying meaningful signal being communicated so that appropriate behavioral responses can be produced. A key step in this process is a many-to-one mapping that bins communication sounds, perhaps carrying similar 'meanings' or associated with specific behavioral responses, into distinct categories. To accomplish this, the auditory system must generalize over the aforementioned variability in the production and transmission of communication sounds. We previously proposed, based on a model of visual categorization (*Ullman et al., 2002*), a theoretical model that identified distinctive acoustic features that were highly likely to be found across most exemplars of a category and were most contrastive with respect to other categories. Using these 'most informative features (MIFs)', the model accomplished auditory categorization with high accuracy (*Liu et al., 2019*). In that study, we also showed that features of intermediate lengths, bandwidths, and complexities were typically more informative for auditory categorization. In a second study, we further showed in a guinea pig (GP) animal model that neurons in the superficial layers of the primary auditory cortex (A1) demonstrated call-feature-selective responses and complex receptive fields that were consistent with model-predicted features, providing support for the model at the neurophysiological level (*Montes-Lourido et al., 2021a*). In this study, we investigated whether the feature-based model held true at a behavioral level, by determining whether the model, trained solely using natural GP calls, could predict GP behavioral performance in categorizing both natural calls as well as calls with altered spectral and temporal features.

Studies in a wide range of species have probed the impact of alterations to spectral and temporal cues on call recognition. For example, in humans, it has been shown that speech recognition relies primarily on temporal envelope cues based on experiments that measured recognition performance when subjects were presented with noise-vocoded speech at different spectral resolutions (*Shannon et al., 1995*; *Smith et al., 2002*). However, recognition is also remarkably resilient when the envelope is altered because of tempo changes—for example, word intelligibility is resilient to a large degree of time-compression of speech (*Janse et al., 2003*). Results from other mammalian species are broadly consistent with findings in humans. In gerbils, it has been shown that firing rate patterns of A1 neurons could be used to reliably classify calls that were composed of only four spectral bands (*Ter-Mikaelian et al., 2013*). In GPs, small neuronal populations have been shown to be resistant to such degradations as well (*Aushana et al., 2018*). Slow amplitude modulation cues have been proposed as a critical cue for the neuronal discriminability of calls (*Souffi et al., 2020*), but behaviorally, call identification can be resilient to large changes in these cues. For example, mice can discriminate between calls that have been doubled or halved in length (*Neilans et al., 2014*). This remarkable tolerance to cue variations might be related to the wide range of variations with which calls are produced in different behavioral contexts. For example, for luring female mice and during direct courtship, male mice modify many call parameters including sequence length and complexity (*Chabout et al., 2015*). In contrast, one study in GPs using naturalistic stimuli (a human footstep sound) demonstrated that while GPs discriminated time-compressed sounds from the natural sound on which they were trained, they did not distinguish between time-expanded sounds and natural sounds (*Ojima and Horikawa, 2015*). Along the spectral dimension, mouse call discrimination can be robust to changes in long-term spectra, including moderate frequency shifts and removal of frequency modulations (*Neilans et al., 2014*). Indeed, it has been suggested that the bandwidth of ultrasonic vocalizations is more important for communication than the precise frequency contours of these calls (*Screven and Dent, 2016*). Again, given that mice also modify the spectral features of their calls in a context-dependent manner (*Chabout et al., 2015*), it stands to reason that their perception of call identity is also robust to alterations of spectral features. In GPs using footstep stimuli, animals reliably discriminated sounds subjected to a band-reject filter from the natural sound (*Ojima and Horikawa, 2015*).

Overall, these studies suggest that for calls, in particular, information is encoded at multiple levels. Whereas the specific parameters of a given call utterance might carry rich information about the identity (*Boinski and Mitchell, 1997*; *Miller et al., 2010*; *Gamba et al., 2012*; *Fukushima et al., 2015*) and internal state of the caller as well as social context (*Seyfarth and Cheney, 2003*; *Coye et al., 2016*), call category identity encompasses all these variations. In some behavioral situations, listeners might need to be sensitive to these specific parameter variations—for example, for courtship, female mice have been shown to exhibit a high preference for temporal regularity of male calls (*Perrodin et al., 2020*). But in other situations, animals must and do generalize over this variability to extract call category identity, which is critical for providing an appropriate behavioral response. What mechanisms

enable animals to generalize over this tremendous variability with which calls are heard and how they accomplish call categorization, however, is not well understood.

In this study, based on our earlier modeling and neurophysiological results (*Liu et al., 2019*; *Montes-Lourido et al., 2021a*), we hypothesized that animals can generalize over this production variability and achieve call categorization by detecting features of intermediate complexity within these calls. To test this hypothesis, we trained feature-based models and GPs to classify multiple categories of natural, spectrotemporally rich GP calls. We then tested the categorization performance of both the model and GPs with manipulated versions of the calls. We found that the feature-based model of auditory categorization, trained solely using natural GP calls, could explain about 50% of the overall variance of GP behavioral responses to manipulated calls. By comparing different model versions, we could derive further insight into possible behavioral strategies used by GPs to solve these call categorization tasks. Examining the factors contributing to high model performance in different conditions also provided insight into why a feature-based encoding strategy is highly advantageous. Overall, these results provide support at a behavioral level for a feature-based auditory categorization model, further validating our model as a novel and powerful approach to deconstructing complex auditory behaviors.

## Results

The acquisition of learned categorization behaviors likely involves two underlying processes—the acquisition of the knowledge of auditory categories, and the expression of this knowledge by learning the association between categories and reward outcomes (*Kuchibhotla et al., 2019*; *Moore and Kuchibhotla, 2022*). Many models have been developed to characterize the latter process, that is, the association of a stimulus with reward. Examples include models of classical conditioning and reinforcement learning (*Rescorla and Wagner, 1972*), models that account for the motivational state of subjects during behavior (*Berditchevskaia et al., 2016*), models that account for exploratory drive in subjects (*Pisupati et al., 2021*), or models that account for arousal state (*de Gee et al., 2020*). However, the former process of how knowledge of auditory categories is acquired, especially when categories are 'non-compact' or evident only in multiparametric spaces, is far less understood. Thus, while we briefly describe the behavioral acquisition of a vocalization categorization task for completeness (*Figure 1—figure supplement 1* ), in this study, our goal was to determine the computational principles underlying the former process, that is, what features animals may use to acquire knowledge about acoustic stimulus category, and in particular, vocalization categories.

### GPs learn to report call category in a Go/No-go task

We trained GPs on call categorization tasks using a Go/No-go task structure. Animals initiated trials by moving to the 'home base' region of the behavioral arena (*Figure 1A, B*). Stimuli were presented from an overhead speaker. On hearing Go stimuli, GPs were trained to move to a reward region, where they received a food pellet reward. The correct response to No-go stimuli was to remain in the home base. We trained two cohorts of GPs to categorize two pairs of call categories—Cohort 1 was trained on chuts (Go) versus purrs (No-go), calls that had similar spectral content (long-term spectral power) but different temporal (overall envelope) structure (*Figure 1C*), and Cohort 2 was trained on wheeks (Go) versus whines (No-go), calls that had similar temporal structure but different spectral content (*Figure 1D*). GPs were trained on this task over multiple short sessions every day (~6 sessions of ~40 trials each, ~10 min per session; see Materials and methods). On each trial, we presented a randomly chosen exemplar from an initial training set of 8 exemplars per category. We estimated hit rates and false alarm (FA) rates from all trials in a given day and computed a sensitivity index ($d'$). GPs were considered trained when $d'$ reliably crossed a threshold of 1.5. On average, GPs acquired this task after ~2–3 weeks of training (~4,000 total trials,~250 trials per exemplar; *Figure 1—figure supplement 1*). On the last day of training, GPs displayed a mean $d'$ of 1.94±0.26 for the chuts versus purrs task, and a mean $d'$ of 1.90±0.57 for the wheeks versus whines task.

To gain insight into possible behavioral strategies that GPs might adopt to solve the categorization task, we examined trends of behavioral performance over the training period. Initially, GPs exhibited low hit rates as well as low FA rates, suggesting that they did not associate the auditory stimulus with reward (*Figure 1—figure supplement 1D*). Note that this initial phase was not recorded for the

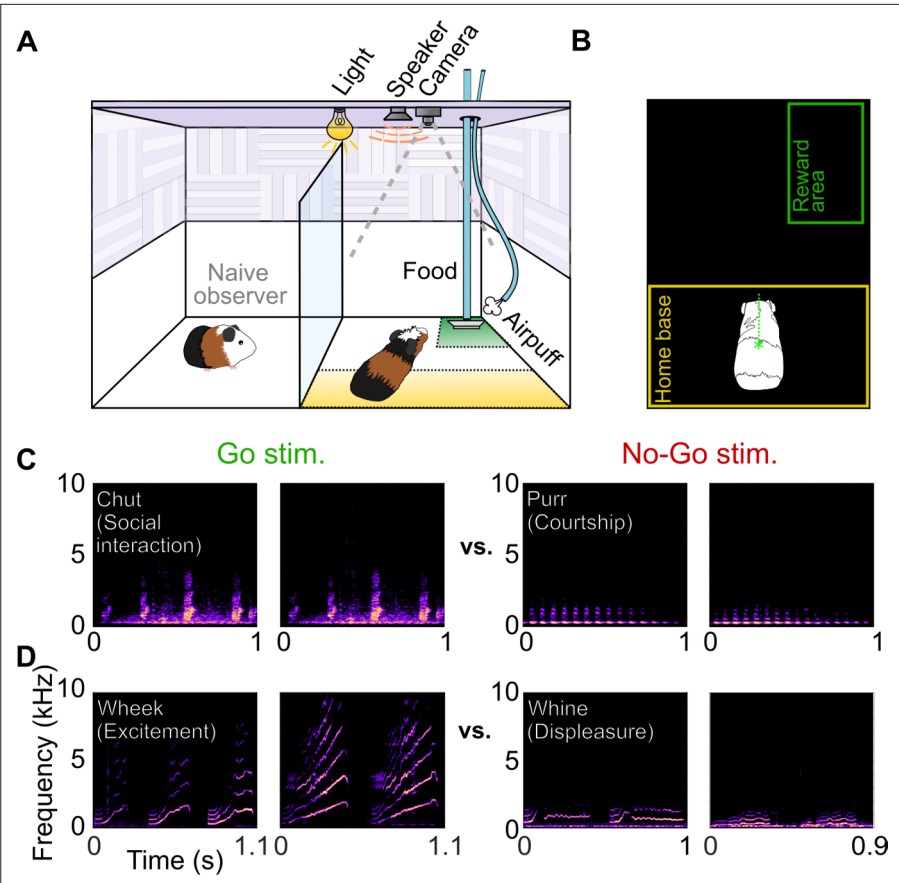

**Figure 1.** Call categorization behavior in GPs. (**A**) Behavioral setup, indicating home base region for trial initiation (yellow) and reward area (green). Some naive animals observed expert animals performing the task to speed up task acquisition. (**B**) Video tracking was employed to detect GP position and trigger task events (stimulus presentation, reward delivery, etc.). (**C**) Spectrograms of example chut calls (Go stimuli for Cohort 1) and purr calls (No-go stimuli for Cohort 1). (**D**) Spectrograms of example wheek calls (Go stimuli for Cohort 2) and whine calls (No-go stimuli for Cohort 2). Performance of GPs during the training phase of the call categorization task is shown in *Figure 1—figure supplement 1*. GP, guinea pig.

The online version of this article includes the following source data and figure supplement(s) for figure 1:

**Figure supplement 1.** Learning rates of GPs performing a call categorization task.

**Figure supplement 1—source data 1.** Hit rates, false alarm rates, and *d′* values of GPs over the course of training on vocalization categorization tasks.

first cohort (chuts vs. purrs task, *Figure 1—figure supplement 1A*). Within 2–3 days, GPs formed a stimulus-reward association and exhibited 'Go' responses for all stimuli but did not discriminate between Go and No-go stimulus categories. This resulted in high hit rates as well as FA rates, but low *d′*. For the remainder of the training period, hit rates remained stable whereas FA rates gradually declined, suggesting that the improvements to *d′* resulted from GPs learning to suppress responses to No-go stimuli (*Figure 1—figure supplement 1A, B, D, E*).

While these data were averaged over all sessions daily for further analyses, we noticed within-in-day trends in performance that might provide insight into the behavioral state of the GPs. We analyzed performance across intra-day sessions, averaged over 4 days after the animals acquired the task (*Figure 1—figure supplement 1C, F*). In early sessions, both hit rates and FA rates were high, suggesting that the GPs weighted the food reward highly, risking punishments (air puffs/time outs) in the process. In subsequent sessions, both the hit rate and FA rate declined, suggesting that the GPs shifted to a punishment-avoidance strategy. Despite these possible changes in decision criteria used by the GPs, they maintained consistent performance, as *d′* remained consistent across sessions. Therefore, in all further analyses, we used *d′* values averaged over all sessions as a performance metric.

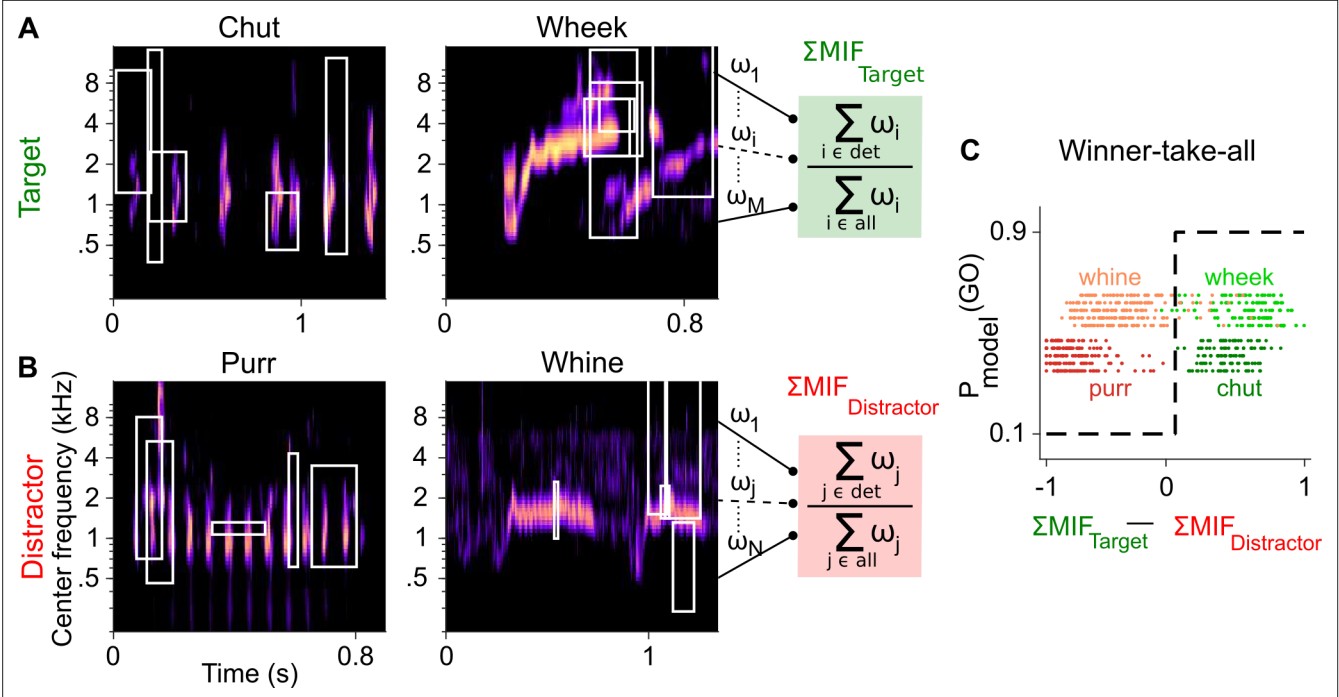

**Figure 2.** Framework of the model trained to perform call categorization tasks. (**A**) and (**B**) Example cochleagrams for target (**A**) and distractor (**B**) calls. Cochleagram rows were normalized to set the maximum value as 1 and then smoothed for display. White rectangles denote detected MIFs for that call. For an input call, the target (green) FD stage response is the sum of all detected target MIF weights normalized by the sum of all MIF weights for that call type. The distractor response (red) is similarly computed. (**C**) The output of the winner-take-all (WTA) stage is determined based on the difference between the target and distractor FD stage responses. Dots represent the WTA outputs for all calls used for training the models. Rows represent the five instantiations of the model with different MIF sets. MIF, maximally informative features; det, detected MIFs; all, all MIFs.

Note that these observations of possible motivational and other fluctuations are not intended to be captured by our model. Rather, our model solely focuses on what spectrotemporal features GPs can use to deduce stimulus categories.

## A feature-based computational model can be trained to accomplish call categorization

In parallel, we extended a feature-based model that we previously developed for auditory categorization (*Liu et al., 2019*) to accomplish GP call categorization in a Go/No-go framework. Briefly, we implemented a three-layer model consisting of a spectrotemporal representation layer, a feature-detection (FD) layer, and a winner-take-all (WTA) layer. The spectrotemporal layer was a biophysically realistic model of the auditory periphery (*Zilany et al., 2014*). For the FD layer, we used greedy search optimization and information theoretic principles to derive a set of MIFs for each call type that was optimal for the categorization of that call type from all other call types (*Figure 2A, B*; *Liu et al., 2019*). We derived five distinct sets of MIFs for each call type that could accomplish categorization (see

**Table 1.** Properties of MIFs.

| Call name | Instantiation | Number of MIFs | MIF duration (ms) (mean±std) | MIF bandwidth (octaves) (mean±std) |
|---|---|---|---|---|
| Chut | 1, 2, 3, 4, 5 | 20, 20, 20, 20, 20 | 88±63, 106±53, 108±56, 109±64, 133±47 | 4.0±2.0, 4.4±1.2, 3.1±1.9, 3.7±1.9, 2.6±1.8 |
| Purr | 1, 2, 3, 4, 5 | 8, 9, 20, 20, 20 | 91±49, 83±43, 116±49, 116±56, 86±63 | 2.6±1.2, 2.8±1.2, 3.1±1.4, 3.2±1.5, 3.6±1.2 |
| Wheek | 1, 2, 3, 4, 5 | 8, 14, 13, 11, 12 | 144±47, 99±58, 104±68, 116±62, 114±65 | 2.3±1.6, 2.6±1.8, 2.9±2.2, 2.1±1.1, 2.5±1.7 |
| Whine | 1, 2, 3, 4, 5 | 20, 20, 15, 20, 20 | 109±55, 111±68, 133±37, 117±51, 108±70 | 3.5±1.8, 3.4±1.6, 2.6±1.4, 3.2±1.5, 3.9±1.6 |
| Summary | | 16.5±4.7 | 109±57 | 3.2±1.7 |

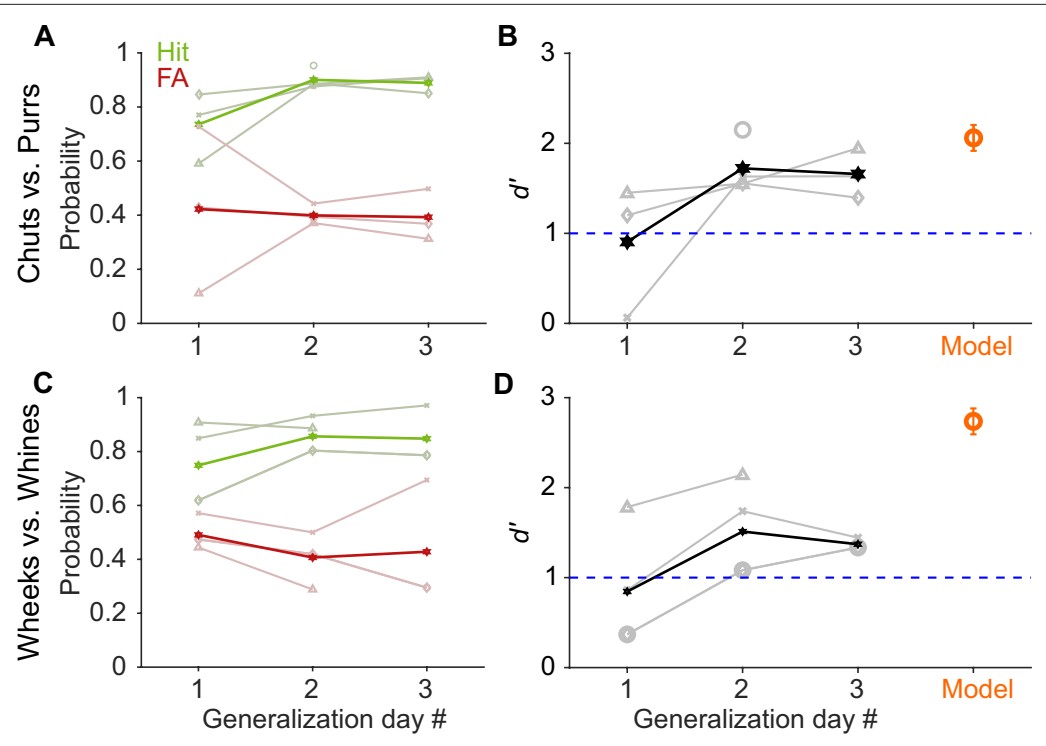

**Figure 3.** GP and model performance generalizes to new exemplars. (**A**) and (**C**) Hit (green) and false alarm (FA; red) rates of GPs when categorizing new exemplars as a function of generalization day. We presented ~5 trials of each new exemplar per day. Dark lines correspond to average over subjects, faint lines correspond to individual subjects. (**B**) and (**D**) Quantification of generalization performance. Black line corresponds to average $d'$, gray lines are $d'$ values of individual subjects. GPs achieved a $d'>1$ by generalization day 2, that is, after exposure to only ~5 trials of each new exemplar on day 1. The feature-based model (orange, n = 5 instantiations, error bars correspond to s.e.m.) also generalized to new exemplars that were not part of the model's training set of calls. Source data are available in *Figure 3—source data 1*. GP, guinea pig.

The online version of this article includes the following source data for figure 3:

**Source data 1.** Hit rates, false alarm rates, and $d'$ values of GPs and the feature-based model for generalization to new vocalization exemplars.

---

Materials and methods). We refer to models using these distinct MIF sets as different instantiations of the model.

Call-specific MIF sets in the FD layer showed near-perfect performance (area under the curve, or AUC>0.97 for all 20 MIF sets [4 call categories×5 instantiations per category], mean=0.994) in categorizing target GP calls from other calls in the training data set. Similar to results from *Liu et al., 2019*, the number of MIFs for each instantiation of the model ranged from 8 to 20 (mean=16.5), with MIFs spanning ~3 octaves in bandwidth and ~110 ms in duration on average (*Table 1*). To assess the performance of the WTA layer based on these training data, we estimated $d'$ using *equation 1* (Materials and methods). The WTA output also showed near-perfect performance for classifying the target from the distractor for both chuts versus purrs (mean $d'$=4.65) and wheeks versus whines (mean $d'$=3.69) tasks (*Figure 2C*).

## Both GPs and the model generalize to new exemplars

To determine if GPs learned to report call category or if they simply remembered the specific call exemplars on which they were trained, we tested whether their performance generalized to a new set of Go and No-go stimuli (eight exemplars each) that the GPs had not encountered before. On each generalization day, we ran four sessions of ~40 trials each, with the first two sessions containing only training exemplars and the last two sessions containing only new exemplars. All GPs achieved a high-performance level ($d'>1$) to the new exemplars by generalization day 2 (*Figure 3*), that is, after being

exposed to only a few repetitions of the new exemplars (~5 trials per new exemplar on generalization day 1). As an additional control to ensure that GPs did not rapidly learn reward associations for the new exemplars, for GPs performing the wheeks versus whines task (*n*=3), we also quantified generalization performance when the regular training exemplars and a second new set of exemplars were presented in an interleaved manner (400 trials with an 80/20 mix of training and new exemplars). GPs achieved *d'*>1 for new exemplars in this interleaved set as well, further supporting the notion that GPs were truly reporting call category.

Similar to GPs, to test model generalization, we quantified model performance for new call exemplars (*Figure 3B, D*). Models using different MIF sets, that is, all instantiations of the model for chut, purr, wheek, and whine classification achieved high categorization performance (*d'*>1) for the new

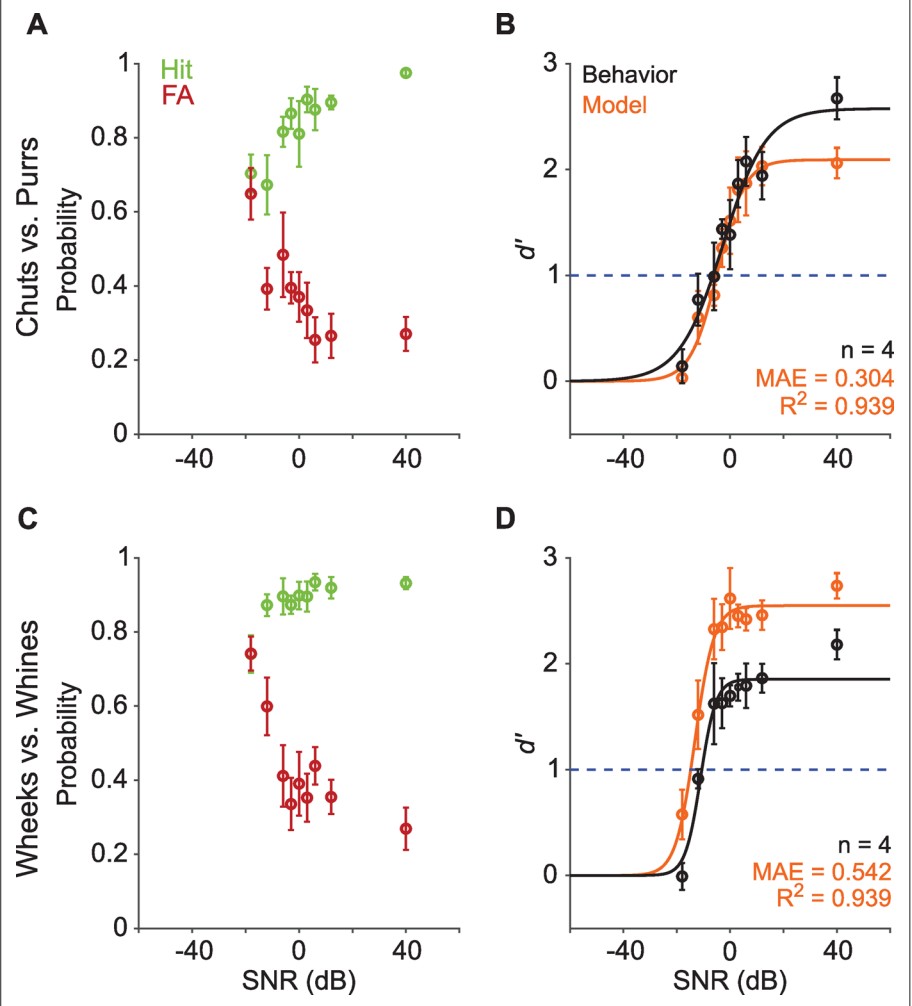

**Figure 4.** Call categorization is robust to degradation by noise. (**A**) and (**C**) Hit (green) and false alarm (FA; red) rates of GPs categorizing calls with additive white noise at different SNRs. (**B**) and (**D**) Sensitivity index (*d'*) as a function of SNR. Black symbols correspond to the mean *d'* across animals (*n*=4); error bars correspond to s.e.m. Black line corresponds to a psychometric function fit to the behavioral data. Orange symbols correspond to the mean *d'* across five instantiations of the model, error bars correspond to s.e.m. Orange line corresponds to a psychometric function fit to the model data. Dashed blue line signifies *d'*=1. The statistical significance of SNR value on behavior was evaluated using a generalized linear model (see main text). Source data are available in *Figure 4—source data 1*. GP, guinea pig; MAE, mean absolute error; SNR, signal-to-noise ratio.

The online version of this article includes the following source data for figure 4:

**Source data 1.** Hit rates, false alarm rates, and *d'* values of GPs and the feature-based model for vocalization categorization in noise (at different SNRs).

exemplars. In summary, GPs as well as the feature-based model could rapidly generalize to novel exemplars.

## Both GPs and the model exhibit similar categorization-in-noise thresholds

Real-world communication typically occurs in noisy listening environments. To test how well GPs could maintain categorization in background noise, we assessed their performance when call stimuli were masked by additive white noise at several signal-to-noise ratios (SNRs) for both Go and No-go stimuli. Experiments were conducted in a block design, using a fixed SNR level per session (~40 trials) and testing 5 or 6 SNR levels each day. At the most favorable SNR (>20 dB), GPs exhibited high hit rates and low FA rates, leading to high $d'$ (>2) for both call groups (*Figure 4*). With increasing noise level (i.e., decreasing SNR), we observed a decrease in hit rate and an increase in FA, as expected, with a concomitant decrease in $d'$. To determine the effect of SNR value on $d'$, we constructed a generalized linear model with a logit link function to predict trial-by-trial behavioral outcomes, with stimulus type (Go or No-go), SNR value, and an interaction term as predictors, and animal ID as a random effect (see *Equation 3*, Materials and methods). We compared this full model to a null model consisting only of stimulus type as a predictor and animal ID as a random effect (see *Equation 5*, Materials and methods). This comparison revealed a strong effect of SNR value of $d'$ (chuts vs. purrs: $\chi^2$=124.6, $p$=2.2×10$^{-16}$; wheeks vs. whines: $\chi^2$=182.5; $p$=2.2×10$^{-16}$). At the most adverse SNR (–18 dB) for both call groups, hit and FA rates were similar, suggesting that the animals were performing at chance level. To estimate the SNR corresponding to the performance threshold ($d'$=1) for call categorization in noise, we fit a psychometric function to the behavioral $d'$ data (see Materials and methods). We obtained performance thresholds (SNR at which $d'$=1) for both the chuts versus purrs (–6.8 dB SNR) and wheeks versus whines (–11 dB SNR) tasks that were qualitatively similar to human speech discrimination performance in white noise (*Phatak and Allen, 2007*).

We also tested the performance of the feature-based model (trained only on clean stimuli) on the same set of noisy stimuli as the behavioral paradigm. Model performance trends mirrored behavior, with a higher threshold for the chuts vs. purrs task (–5.4 dB SNR) compared to the wheeks versus whines task (–15 dB SNR). Similar to behavioral data, we confirmed a significant effect of SNR value on model performance using a trialwise GLM analysis (chuts vs. purrs: $\chi^2$=402.3, p=2.2×10$^{-16}$; wheeks vs. whines: $\chi^2$=203.7; p=2.2×10$^{-16}$). To compare behavioral data with the model, we fit a line to the $d'$ values obtained from the model plotted against the $d'$ values obtained from behavior, and computed the $R^2$ value. We used the mean absolute error (MAE) to quantify the absolute deviation between model and behavior $d'$ values. Although the model over-performed for the wheeks vs. whines task, it could explain a high degree of variance ($R^2$=0.94 for both tasks) of GP call-in-noise categorization behavior.

## Stimulus information might be available to GPs in short-duration segments of calls

Several studies across species, including humans (*Marslen-Wilson and Zwitserlood, 1989*; *Salasoo and Pisoni, 1985*), birds (*Knudsen and Gentner, 2010*; *Toarmino et al., 2011*), sea-lions (*Pitcher et al., 2012*), and mice (*Holfoth et al., 2014*), have suggested that the initial parts of calls might be the most critical parts for recognition. We reasoned that if that were the case for GPs as well, and later call segments did not add much information for call categorization, we might observe a plateauing of behavioral performance after a certain length of a call was presented. To test this, we presented call segments of different lengths (50–800 ms) beginning at the call onsets (*Figure 5A, D*) to estimate the minimum call duration required for successful categorization by GPs. Trials were presented in a randomized manner in sessions of ~40 trials, that is, each trial could be a Go or No-go stimulus of any segment length. We did not observe systematic changes to $d'$ values when comparing the first and second halves of the entire set of trials used for testing, demonstrating that the GPs were not learning the specific manipulated exemplars that we presented. GPs showed $d'$ values >1 for as small as 75 ms segments for both tasks, and as expected, the performance stabilized for all longer segment lengths (*Figure 5B, C, E and F*). As with the SNR experiment, we used a comparison between a full GLM including stimulus length value and a null GLM to evaluate the statistical significance of the effect of stimulus length on behavioral performance. We confirmed a significant effect of segment length

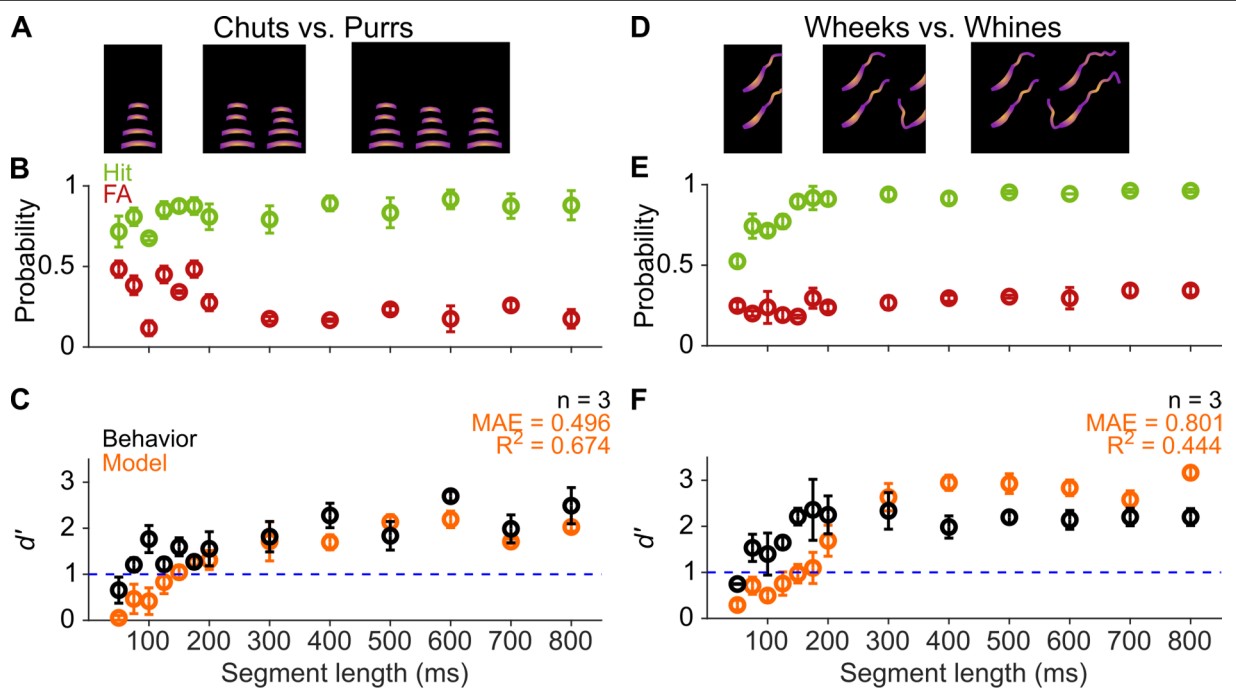

**Figure 5.** GPs can obtain information for categorization from short-duration segments of calls. (**A**) and (**D**) Schematic showing truncation of stimuli at different segment lengths from the onset of calls. (**B**) and (**E**) Average (*n*=3 GPs) hit rate (green) and false alarm rates (red) as a function of stimulus segment length. (**C**) and (**F**) Black symbols correspond to average GP *d'* (*n*=3 GPs), error bars correspond to s.e.m. Orange symbols correspond to average model *d'* (*n*=5 model instantiations), error bars correspond to s.e.m. Dashed blue line denotes *d'*=1. The statistical significance of segment length value on behavior was evaluated using a generalized linear model (see main text). Source data are available in *Figure 5—source data 1*. FA, false alarm; GP, guinea pig.

The online version of this article includes the following source data for figure 5:

**Source data 1.** Hit rates, false alarm rates, and *d'* values of GPs and the feature-based model for the segment-length manipulation.

on behavioral performance (chuts vs. purrs: $\chi^2$=58.7, *p*=1.8×10⁻¹³; wheeks vs. whines: $\chi^2$=120.2; *p*=2.2×10⁻¹⁶). These data suggest that short-duration segments of calls carry sufficient information for call categorization, at least in the tested one-vs.-one scenarios. The fact that call category can be extracted from the earliest occurrences of such segments suggests two possibilities: (1) a large degree of redundancy is present in calls, or (2) the repeated segments can be used to derive information beyond call category (e.g., caller identity or emotional valence).

Model performance, however, only exceeded a *d'* value of 1 for ~150 ms call segments, and performance only plateaued after a 200 ms duration (*Figure 5C, F*). The effect of segment length on model performance was statistically significant (chuts vs. purrs: $\chi^2$=613.2, *p*=1.8× 0⁻¹³; wheeks vs. whines: $\chi^2$=775.4; *p*=2.2×10⁻¹⁶). This observation could reflect the fact that the MIFs identified for categorization were on average about 110 ms long. Despite these differences, model performance was in general agreement with behavioral performance for both the chuts versus purrs and wheeks versus whines tasks ($R^2$=0.674 and 0.444, respectively).

## Temporal manipulations had little effect on model performance and GP behavior

To investigate the importance of temporal cues for GP call categorization, we introduced several gross temporal manipulations to the calls. We first started by changing the tempo of the calls, that is, stretching/compressing the calls without introducing alterations to the long-term spectra of calls (*Figure 6A, D*). This resulted in calls that were ~0.45, 0.5, ~0.56, ~0.63, ~0.77, ~1.43, 2.5, and 5 times the original lengths of the calls. As earlier, we presented stimuli in randomized order and verified that *d'* did not vary systematically between the first and second half of trials, suggesting that the GPs were not learning new associations for the manipulated exemplars. GP behavioral performance remained

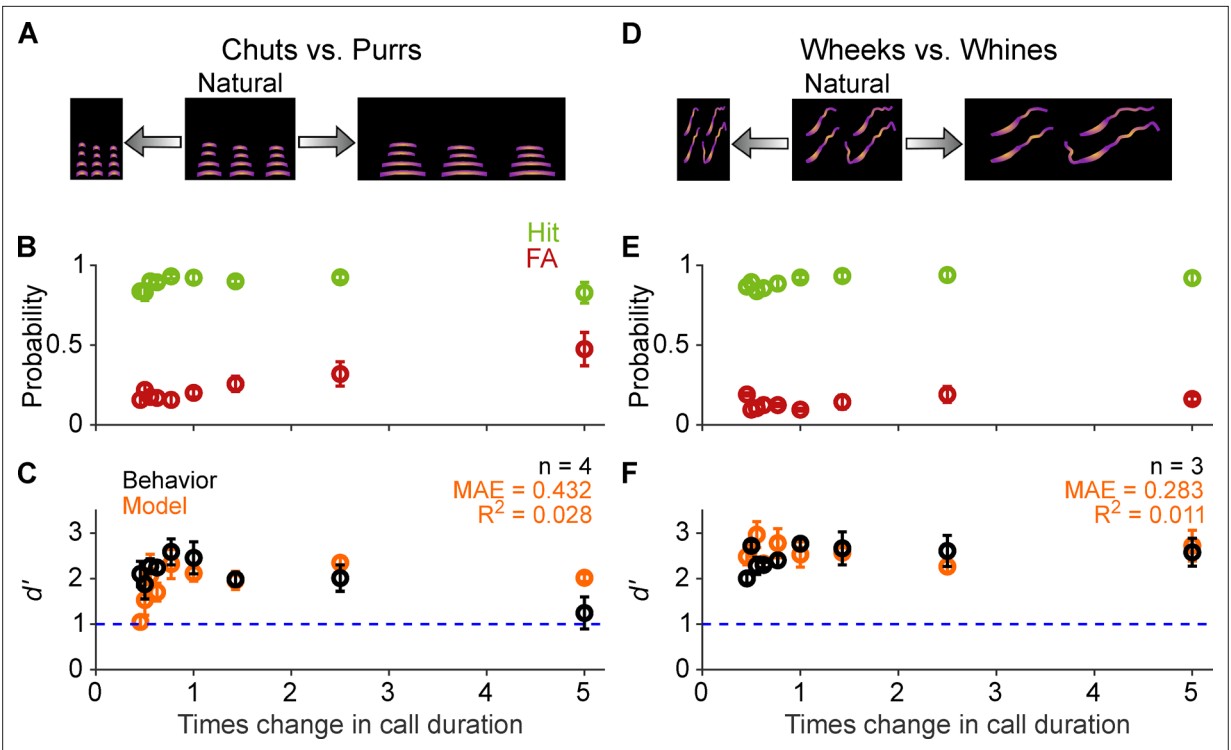

**Figure 6.** Call categorization is resistant to changes in tempo. (**A**) and (**D**) Schematic showing changes to call tempo without affecting spectral content. (**B**) and (**E**) Average (n=4 GPs for chuts vs. purrs; n=3 GPs for wheeks vs. whines) hit rate (green) and false alarm rates (red) as a function of tempo change, expressed as times change in call duration (1 corresponds to the natural call). (**C**) and (**F**) Black points correspond to average GP d'; error bars correspond to s.e.m. Orange points correspond to average model d' (n=5 model instantiations), error bars correspond to s.e.m. Dashed blue line denotes d'=1. The statistical significance of call duration value on behavior was evaluated using a generalized linear model (see main text). Source data are available in *Figure 6—source data 1*. FA, false alarm; GP, guinea pig.

The online version of this article includes the following source data for figure 6:

**Source data 1.** Hit rates, false alarm rates, and d' values of GPs and the feature-based model for tempo-shifted vocalizations.

above a d' of 1 for all these perturbations, showing high hit rates and low FA rates (*Figure 6B, E*) leading to similar d' across probed conditions (*Figure 6C, F*). Similar trialwise GLM analysis as earlier, but including the sign of the manipulation (compression vs. expansion) as an additional predictor in the full model, revealed a weak effect (based on the value of the $\chi^2$ statistic, which roughly corresponds to effect size) of tempo shift on performance (chuts vs. purrs: $\chi^2$=46; p=5.7×10$^{-10}$; wheeks vs. whines: $\chi^2$=13.6; p=0.003).

Similarly, model performance also remained above d' of 1 for all tempo manipulations. We emphasize that the MIFs used in the model were trained only on natural calls, and were not temporally compressed or stretched to match the test stimuli. GLM analysis as above revealed a weak effect of tempo shift on model d' for chuts versus purrs ($\chi^2$=43.3; p=2.14×10$^{-9}$), and no effect for wheeks versus whines ($\chi^2$=0.322; p=0.96). Note that while the model qualitatively captured GP behavioral trends, we obtained low $R^2$ values likely because of the weak effect of tempo shift values on both animal and model behaviors. The variability of the observed behavior was likely dominated by non-stimulus-related factors such as motivation, exploratory drive, or arousal (*Berditchevskaia et al., 2016*; *de Gee et al., 2020*; *Pisupati et al., 2021*), factors not explicitly modeled here. However, the relatively low MAE for the tempo manipulations (comparable with MAEs of the SNR manipulation which showed high $R^2$ values) suggests a broad correspondence between the model and behavior.

The tempo manipulations lengthened or shortened both syllables and inter-syllable intervals (ISIs). Because a recent study in mice (*Perrodin et al., 2020*) suggested that the regularity of ISI values might be crucial for the detection of male courtship songs by female mice, we next asked whether GPs used individual syllables or temporal patterns of calls for call categorization. First, as a low-level control, we replaced the ISIs of calls with silence instead of the low level of background noise present

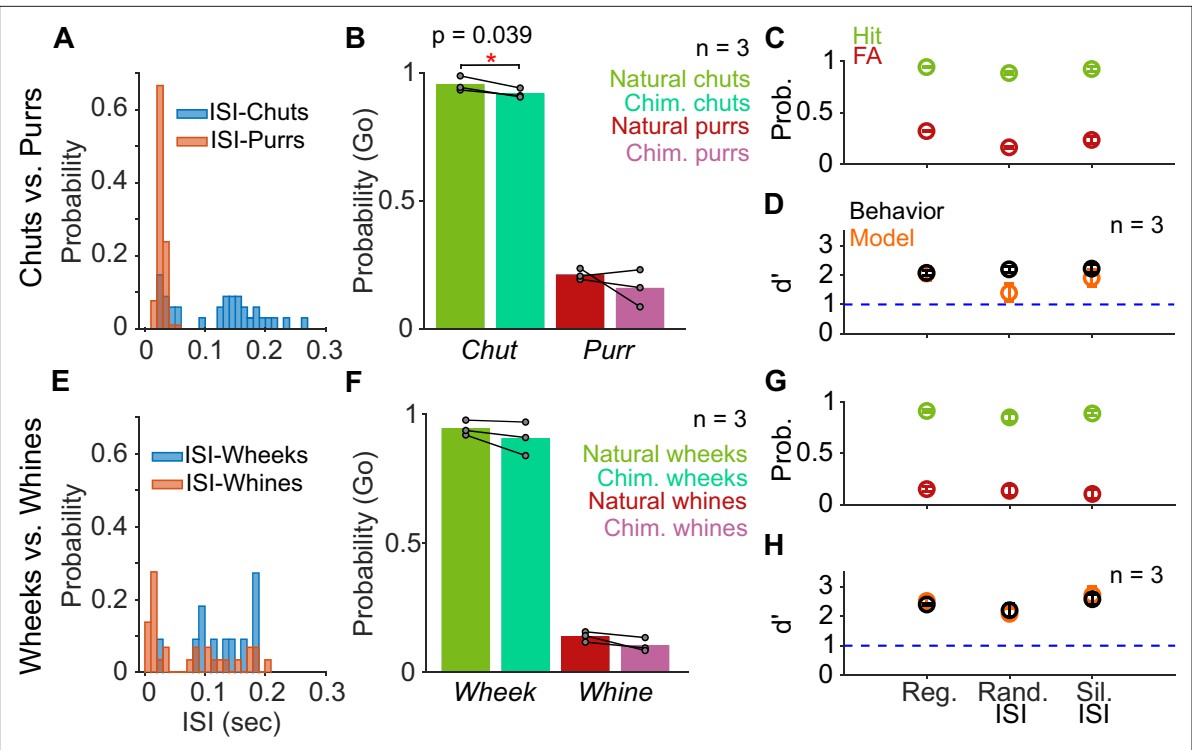

**Figure 7.** Call categorization is resistant to manipulations to the inter-syllable interval (ISI). (**A**) and (**E**) Distribution of ISI lengths for the call types used in the categorization tasks. (**B**) and (**F**) Comparison of the Go rates for natural and chimeric calls. We compared Go rates rather than *d'* because chimeric calls were presented in a catch-trial design (see main text and Materials and methods). Chim. refers to chimeric calls with one call's syllables and the other call's ISIs. For example, chimeric chuts have chut syllables and purr ISIs. Label on x-axis refers to syllable identity. (**C**) and (**G**) Comparison of hit (green) and FA (red) rates for regular calls, calls where we replaced ISI values with values drawn from the same calls' ISI distributions, and calls where we replaced the ISI with silence (rather than background noise). (**D**) and (**H**) Comparison of GP (black; *n*=3 GPs) and model (orange; *n*=5 instantiations) *d'* values across these manipulations. Error bars correspond to s.e.m. Source data are available in *Figure 7—source data 1*. FA, false alarm; GP, guinea pig.

The online version of this article includes the following source data for figure 7:

**Source data 1.** Hit rates, false alarm rates, and *d'* values of GPs and the feature-based model for various ISI manipulations (silent ISI, random ISI, chimeric calls).

in recordings to ensure that GPs were not depending on any residual ISI information (silent ISI). Second, since many call categories show a distribution of ISI durations (*Figure 7A, E*), we replaced the ISI durations in a call with ISI values randomly sampled from the ISI distribution of the same call category (random ISI). The hit and FA rates for both silent and random ISI stimuli were comparable to the regular calls for both categorization tasks (*Figure 7C, G*), and thus, no significant difference in *d'* values was observed across these conditions (*Figure 7D, H*; repeated measures ANOVA; *p*=0.536 for chuts vs. purrs and *p*=0.365 for wheeks vs. whines). Consistent with these behavioral trends, model performance was also largely unaffected by these ISI manipulations.

Finally, because the Go/No-go stimulus categories vary in their ISI distributions (*Figure 7A, E*), particularly for chuts versus purrs, we generated chimeric calls with syllables of one category and ISI values of the other category (e.g., chut syllables with purr ISIs). Since we combined properties of two call categories, we presented chimeric stimuli in a catch-trial design (see Materials and methods) and compared the Go response rates using syllable identity as the label for a category. While the response rates were marginally lower for the chimeric chuts (chut syllables with purr ISI values) compared to regular chuts (paired *t*-test; *p*=0.039), responses were unaltered for regular and chimeric purrs (paired *t*-test; *p*=0.415), chimeric wheeks (paired *t*-test; *p*=0.218), and chimeric whines (paired *t*-test; *p*=0.099) (*Figure 7B, F*). We did not test the model with chimeric stimuli because 'Go' and 'Nogo' category labels could not be assigned.

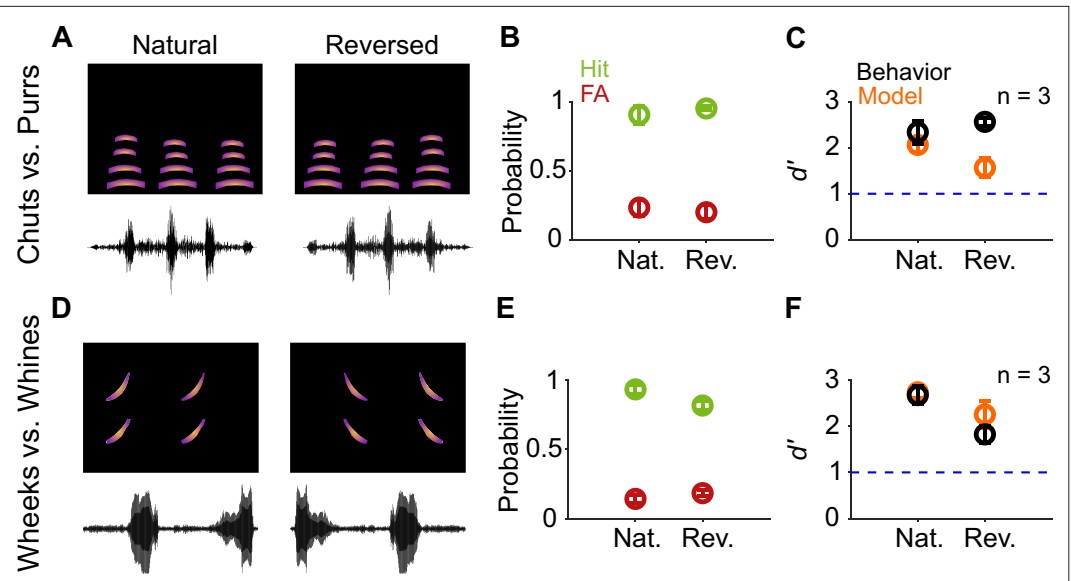

**Figure 8.** Call categorization is resistant to time-reversal. (**A**) and (**D**) Schematics showing spectrogram and waveform of natural (left) and reversed (right) purr (**A**) and wheek (**D**) calls. (**B**) and (**E**) Average (*n*=3 GPs) hit rate (green) and FA rate (red) for natural and reversed calls. (**C**) and (**F**) Average performance of GPs (black; *n*=3 GPs) and model (orange; *n*=5 model instantiations) for natural and reversed calls. Error bars correspond to s.e.m. Source data are available in *Figure 8—source data 1*. FA, false alarm; GP, guinea pig.

The online version of this article includes the following source data for figure 8:

**Source data 1.** Hit rates, false alarm rates, and *d'* values of GPs and the feature-based model for natural and reversed vocalizations.

As a more drastic manipulation, we tested the effects of temporally reversing the calls (*Figure 8A, D*). Given that both chuts and purrs are calls with temporally symmetric spectrotemporal features, compared to natural calls, we observed no changes in the hit and FA rates (*Figure 8B*) or *d'* values for reversed calls (*Figure 8C*; paired *t*-test; *p*=0.582). Wheeks and whines, however, show strongly asymmetric spectrotemporal features. Interestingly, reversal did not significantly affect the categorization performance for this task as well (*Figure 8E, F*; paired *t*-test; *p*=0.151). The model also maintained robust performance (*d'*>1) for call reversal conditions but with an ~18% decrease in *d'* compared to behavior. Overall, these results suggest that GP behavioral performance is tolerant to temporal manipulations such as tempo changes, ISI manipulations, and call reversal, and this tolerance can be largely captured by the feature-based model.

## Spectral manipulations cause similar degradation in model performance and GP behavior

Because temporal manipulations did not significantly affect GP behavioral or model classification performance, we reasoned that categorization was primarily driven by within-syllable spectral cues. To ascertain the impact of spectral manipulations on call categorization, we varied the fundamental frequency (F0) of the calls from one octave lower (–50%) to one octave higher (+100%) than the regular calls without altering call lengths (*Figure 9A and D*). As earlier, we verified that *d'* did not vary systematically between the first and second half of trials, suggesting that the GPs were not learning new associations for the manipulated exemplars. Both increases and decreases to the F0 of the calls significantly affected behavioral performance, revealed by a GLM analysis as described earlier (chuts vs. purrs: $\chi^2$=106.7; *p*=2.2×10$^{-16}$; wheeks vs. whines: $\chi^2$=158.2; *p*=2.2×10$^{-16}$). Particularly, we saw a rise in FA rates (*Figure 9B*) as the F0 deviated farther from the natural values, leading to a significant drop in *d'* values for several conditions (*Figure 9C*). Model performance was also significantly affected by F0 shift magnitude (chuts vs. purrs: $\chi^2$=93.9; *p*=2.2×10$^{-16}$; wheeks vs. whines: $\chi^2$=456; *p*=2.2×10$^{-16}$). As earlier, MIFs used in the model were trained only on natural calls, and were not frequency-shifted to

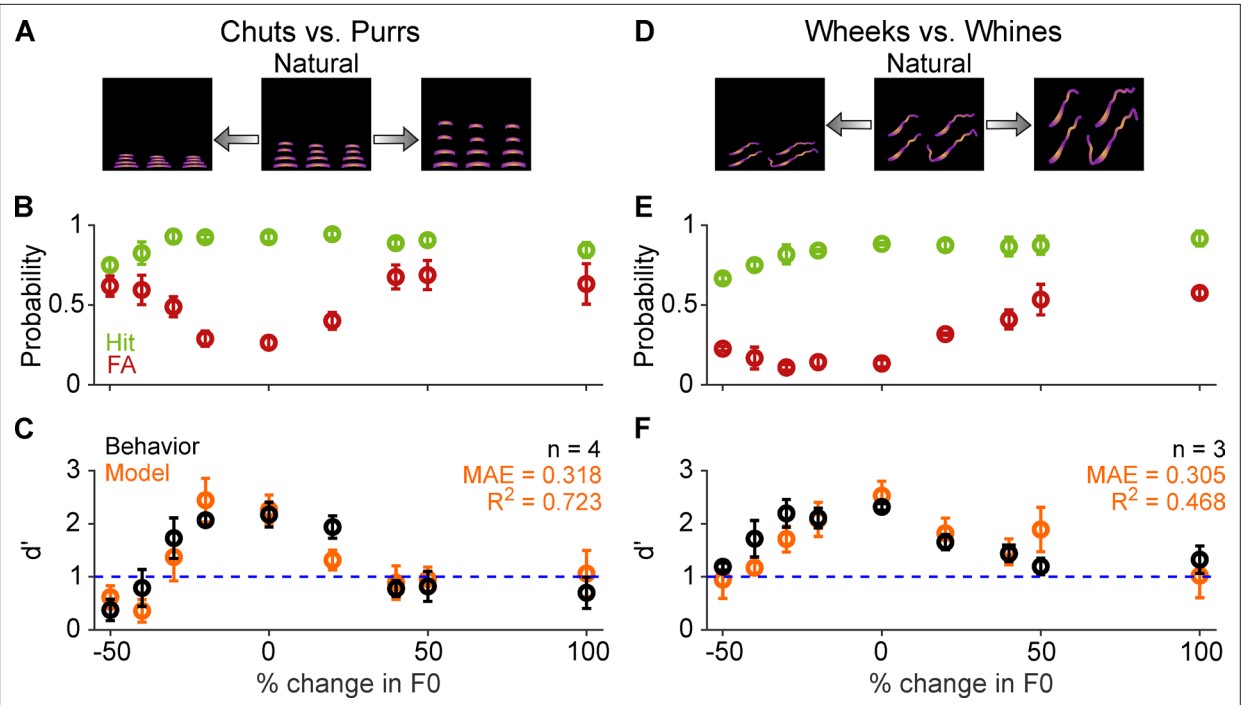

**Figure 9.** Call categorization is sensitive to fundamental frequency (F0) shifts. (**A**) and (**D**) Schematics showing spectrograms of natural calls (middle) and versions where the F0 has been decreased (left) or increased (right). (**B**) and (**E**) Average (n=4 GPs for chuts vs. purrs; n=3 GPs for wheeks vs. whines) hit rate (green) and FA rate (red) for F0-shifted calls Note that 0% change in F0 is the natural call, −50% change corresponds to shifting F0 one octave lower, and 100% change corresponds to shifting F0 one octave higher than the natural call. (**C**) and (**F**) Average performance of GPs (black) and model (orange; n=5 model instantiations) for natural and F0-shifted calls. Error bars correspond to s.e.m. The statistical significance of F0-shift value on behavior was evaluated using a generalized linear model (see main text). Source data are available in *Figure 9—source data 1*. FA, false alarm; GP, guinea pig.

The online version of this article includes the following source data for figure 9:

**Source data 1.** Hit rates, false alarm rates, and *d'* values of GPs and the feature-based model for F0-shifted vocalizations.

match the test stimuli. Model performance mirrored behavioral trends as evidenced by high $R^2$ (0.723 and 0.468 for chuts vs. purrs and wheeks vs. whines, respectively) and low MAE values.

Finally, because wheeks and whines differ in their spectral content at high frequencies (*Figure 1D*), we asked whether GPs exclusively used the higher harmonics of wheeks to accomplish the categorization task. To answer this question, we low-pass filtered both wheeks and whines at 3 kHz (*Figure 10A*), removing the higher harmonics of the wheeks while leaving the fundamental relatively unaffected. Although GP performance showed a decreasing trend for the filtered calls (*Figure 10B, C*), it was not significantly different from regular calls (paired *t*-test; *p*=0.169), indicating that the higher harmonics might be an important but not the sole cue used by GPs for the task. Similar to behavior, the model performed slightly poorly but above a *d'* of 1 in the low-pass filtered condition.

## Feature-based model explains a high degree of variance in GP behavior

The feature-based model was developed purely based on theoretical principles, made minimal assumptions, was trained only on natural GP calls, and had no access to GP behavioral data. For training the model, we used exemplars that clearly provided net evidence for the presence of one category or the other (*Figure 3C*; green and red tick marks in *Figure 11A, D*). We tested the model (and GPs), however, with manipulated stimuli that spanned a large range of net evidence values (histograms in *Figure 11A, D*), with many stimuli close to the decision boundary (blue ticks correspond to an SNR value of −18 dB). Despite the difficulty imposed by this wide range of manipulations, the model explained a high degree of variance in GP behavior as evidenced by high $R^2$ and low MAE across individual paradigms (call manipulations) as well as overall (*Figure 11B–F*; $R^2$=0.60 for chuts vs. purrs and 0.37 for wheeks vs. whines; using model and behavior *d'* values pooled across all tasks).

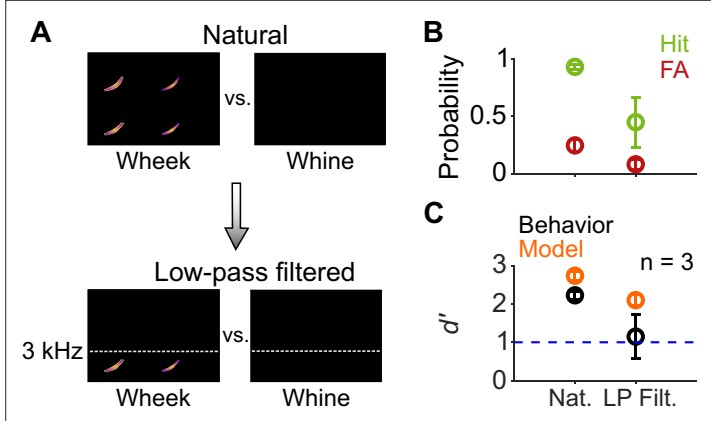

**Figure 10.** Call categorization is mildly affected by low-pass filtering. (**A**) Schematic spectrograms of natural calls (top) and low-pass filtered (bottom) wheek and whine calls. (**B**) Average (*n*=3 GPs) hit rate (green) and FA rate (red) for natural and low-pass filtered (cutoff=3 kHz) calls. (**C**) Average performance of GPs (black) and model (orange; *n*=5 model instantiations) for natural and low-pass filtered calls. Error bars correspond to s.e.m. Source data are available in *Figure 10—source data 1*. FA, false alarm; GP, guinea pig.

The online version of this article includes the following source data for figure 10:

**Source data 1.** Hit rates, false alarm rates, and *d'* values of GPs and the feature-based model for natural and low-pass filtered vocalizations.

To determine whether simpler models based on the overall spectral content of vocalizations could explain some of these results, we built a support vector machine (SVM) classifier that attempted to classify vocalizations based on the long-term spectra of calls (see Materials and methods). Similar to the feature-based model, a WTA stage was implemented by comparing the outputs of the target-call SVM model and the distractor-call SVM model for each input call (response=Go if target-SVM output>distractor SVM output). The spectrum-based SVM model achieved high performance on natural vocalizations (*d'*=3.48 for novel chuts vs. purrs and 2.65 for novel wheeks vs. whines), but failed to capture many aspects of GP responses to manipulated calls, such as the robust performance at low SNRs and modulation of performance by F0-shifted calls. Overall, this resulted in a much lower fraction of the variance explained (*Figure 11—figure supplement 1*). These analyses illustrate that although a simpler model might successfully classify natural vocalizations, applying the model to manipulated stimuli reveals the shortcomings of the spectrum-based model in capturing GP behavioral trends, further highlighting the need for using spectrotemporal features for classification.

## Comparing models with different training procedures yields insight into GP behavioral strategy

The high explanatory power of the feature-based model could be leveraged to gain further insight into what information the GPs were using or learning to accomplish these categorization tasks. On the one hand, because GPs are exposed to these call categories from birth, the GPs may simply be employing the features that they have already acquired for call categorization over their lifetimes to solve our specific categorization tasks. The model presented so far is aligned with this possibility—we trained features to categorize one call type from all other call types (one vs. many categorization) and used a large number of call exemplars for training. Alternatively, GPs could be de-novo learning stimulus features that distinguished between the particular Go and No-go exemplars we presented during training. To test this possibility, we re-trained the model only using the eight exemplars of the targets and distractors that we used to train GPs for one versus one categorization. When tested on manipulated calls, the one versus one model typically performed poorly compared to the original one versus many model. Compared to the one versus many model, the one versus one model was less consistent with behavior as indicated by lower $R^2$ (*Figure 11B, E*) and higher MAE values (*Figure 11C, F*). One explanation for this result could be that eight exemplars of each category are insufficient to adequately train the model. But the one versus one model achieved suprathreshold (*d'*>1) performance on both the training and generalization call exemplars, suggesting that these training data

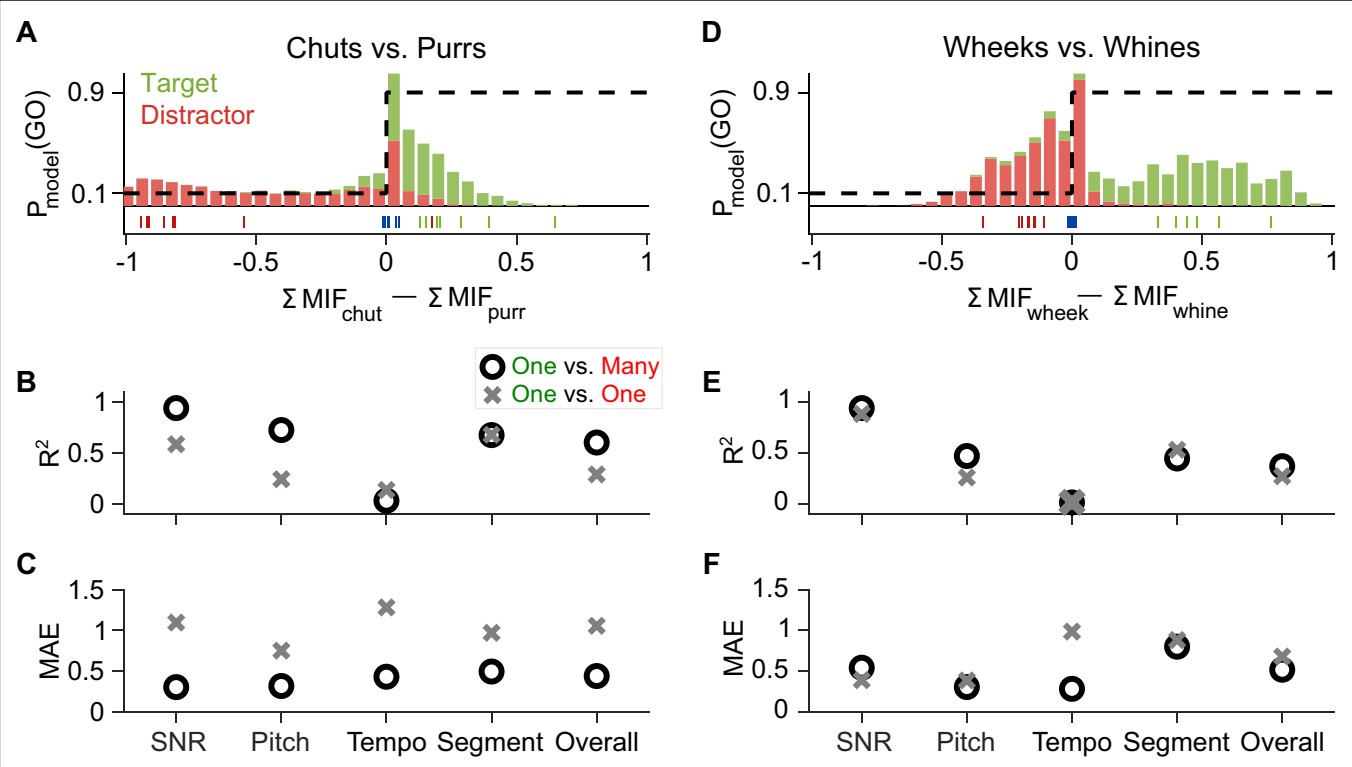

**Figure 11.** Feature-based model explains ~50% of the overall variance in GP behavior. (**A**) Stacked distributions of the evidence for the presence of Go (green) and No-go (red) stimuli (across all manipulations for the chuts vs. purrs task), showing that the output is generally >0 for chuts (green; Go stimulus) and <0 for purrs (red; No-go stimulus). The evidence for easy tasks, such as generalizing to new natural chuts (green ticks) or purrs (red ticks), is typically well away from 0 (decision boundary). In contrast, the evidence for difficult tasks, such as the −18 dB SNR condition (blue ticks), falls near 0. Dashed black line corresponds to the winner-take-all output as a probability of reporting a Go response. (**B, C**) Compared to the model trained with the specific task performed by the GP (chuts vs. purrs; one vs. one), the model trained to classify each call type from all other call types (one vs. many) was more predictive of behavior as indicated by higher $R^2$ (**B**) and lower MAE (**C**). (**D–F**) Same as (**A–C**) but for the wheeks versus whines task. The performance of an SVM classifier that uses the long-term spectrum to classify natural and manipulated calls is shown in ***Figure 11—figure supplement 1***. The performance of a feature-based classifier, with feature duration constrained to 75 ms, is shown in ***Figure 11—figure supplement 2***. GP, guinea pig; MAE, mean absolute error; SVM, support vector machine.

The online version of this article includes the following figure supplement(s) for figure 11:

**Figure supplement 1.** Performance of an SVM classifier on call categorization based on long-term spectrum.

**Figure supplement 2.** Call categorization performance of a feature-based model with feature duration restricted to 75 ms.

were, to some degree, sufficient for classifying natural calls. Furthermore, if GPs were indeed learning stimulus features from the training data set, they would also face the same constraints on training data volume as the model. A second explanation is that the one versus many model better matches GP behavior because rather than re-learning new task-specific features, GPs might be using call features that they had acquired previously over their lifespan to solve our call categorization task. These results also suggest that training a feature-based categorization system (in-silico or in-vivo) on exemplars that capture within-category variability is critical to obtain a system that can flexibly adapt and maintain robust performance to unheard stimuli that exhibit large natural or artificial variations.

The effect of training our model on the one versus many categorization task using a large number of call exemplars for training was that the model learned features that truly captured the within-class and outside-class variability of calls. This resulted in a model that accurately predicted GP performance across a range of stimulus manipulations. To understand how the model was able to achieve robustness to stimulus variations, and to gain insight into how GPs may flexibly weight features differently across the various stimulus manipulations, we examined the relative detection rates of various model MIFs across different stimulus paradigms in which we observed strong behavioral effects (***Figure 12***, ***Figure 12—figure supplement 1***).

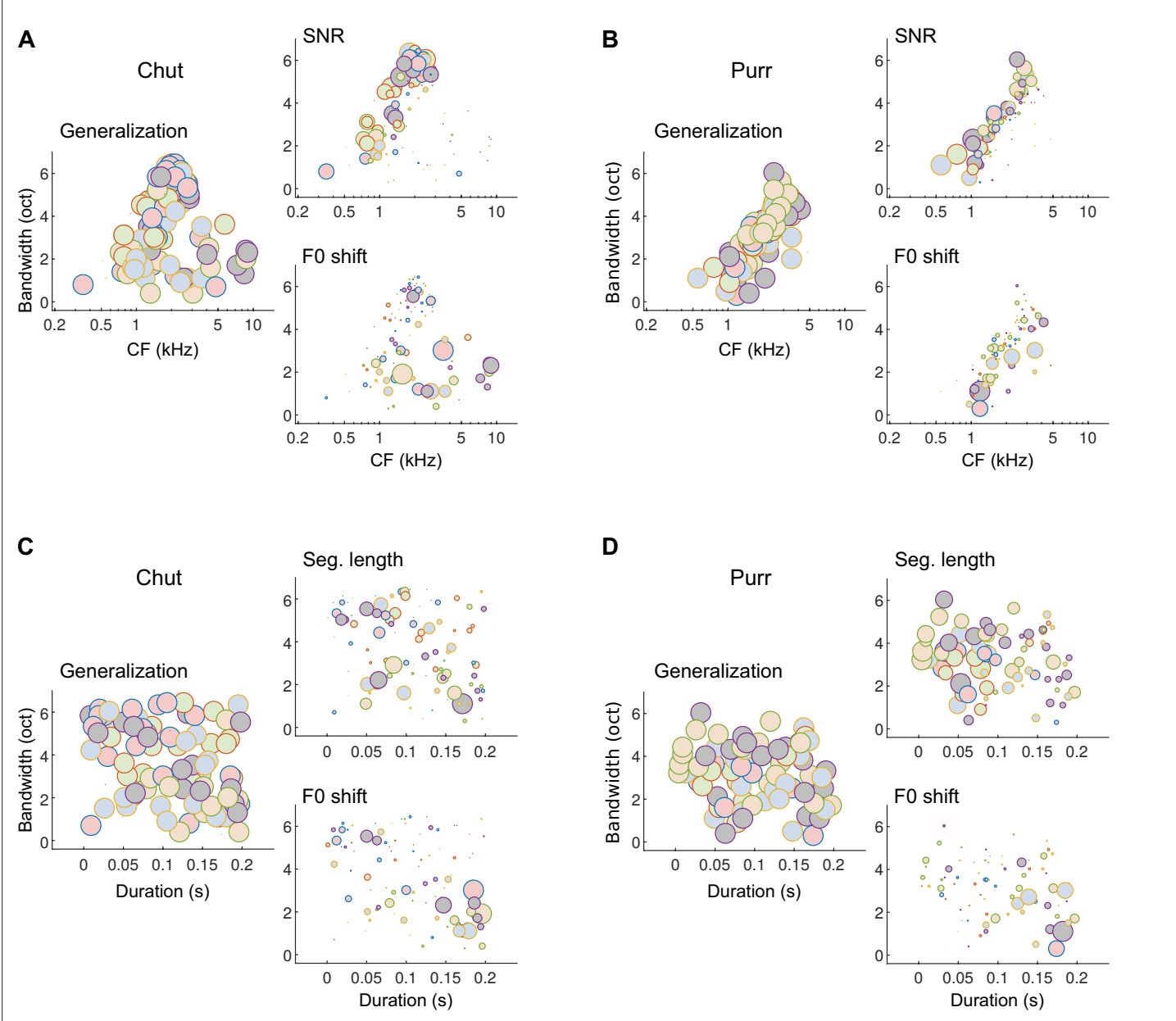

**Figure 12.** Different subsets of MIFs are flexibly recruited to solve categorization tasks for different manipulations. (**A**) We estimated the relative detection rate (i.e., the difference between the detection rate of a given MIF for all within-category and outside-category calls) of all MIFs (discs) for each behavioral paradigm (e.g., SNR). Colors denote different instantiations of the MIFs. Disc diameter is monotonically proportional to the relative detection rate, using a power-law relationship (fourth power) to highlight the most robust features. While MIFs of all center frequencies (CFs) and bandwidths were uniformly recruited for generalizing calls of chut call type, MIFs with lower CFs were preferentially selected for SNR conditions, likely because high-frequency chut features were masked by white noise. In contrast, MIFs with high CF were preferred by the model to solve the F0-shift task. (**B**) Similar results were obtained for purrs. (**C**) MIFs of all durations and bandwidths were uniformly recruited for generalizing calls of chut call type. In contrast, shorter duration MIFs were preferred for segment-length conditions whereas longer-duration MIFs were preferentially recruited for F0-shift conditions. (**D**) Results were similar for purrs. Source data are available in *Figure 12—source data 1*. Similar analyses for wheeks and whines are presented in *Figure 12—figure supplement 1*.

The online version of this article includes the following source data and figure supplement(s) for figure 12:

**Source data 1.** Characteristics of most informative features (CF, bandwidth, and duration) and their relative detection rates for the various stimulus manipulations.

**Figure supplement 1.** Different subsets of MIFs are flexibly recruited to solve categorization tasks for different manipulations (wheeks vs. whines).

In *Figure 12*, we plot the relative detection rates of MIFs (discs, ~20 MIFs per instantiation) from different model instantiations (colors, 5 instantiations) for the different call types used. That is, we computed the difference between the rate of detection of each MIF (for each call category) in response to within-category and outside-category stimuli, and plotted this difference (disc areas) as a function of MIF tuning properties (CF, bandwidth, and duration). For the generalization stimuli, that is, new natural calls on which the model had not been trained, almost all MIFs showed relatively large net detection rates which resulted in plots (*Figure 12A–D*, left panels) with discs of about equal area. For example, all chut MIFs were detected at high rates in novel chut calls (within-category calls), and detected at low rates in non-chut call types (outside-category calls). Note, however, that the learned MIFs are spread out across a range of CFs, bandwidths, and durations. Given these data alone, one might argue that learning ~20 MIFs per call category is highly redundant, and that high performance could be achieved using only a subset of these MIFs. But examining which features maintain high relative detection rates in other stimulus paradigms underscores the utility of learning this wide feature set. When we added white noise to the stimulus, low-CF features showed higher relative detection rates (*Figure 12A, B*, right top) and thus contributed more towards categorization. This could likely be attributed to GP calls having high power at low frequencies, resulting in more favorable local SNRs at lower frequencies. But when we altered stimulus F0, high-CF features contributed more towards categorization (*Figure 12A, B*, right bottom). Similarly, low-duration, high-bandwidth features contributed more when categorizing time-restricted calls, whereas high-duration, low-bandwidth features contributed more when categorizing F0-shifted calls (*Figure 12C, D*). That the model quantitatively matched GP behavior suggests that a similar strategy might be employed by GPs as well. Note that our contention is not that the precise MIFs obtained in our model are also the precise MIFs used by the GPs—indeed, we were able to train several distinct MIF sets that were equally proficient at categorizing calls. Rather, we are proposing a framework in which GPs learn intermediate-complexity features that account for within-category variability and best contrast a call category from all other categories, and similar to the model, recruit different subsets of these features to solve different categorization tasks.

## Discussion

In this study, we trained GPs to report call categories using an appetitive Go/No-go task. We then tested GP call categorization when we challenged them with spectrally and temporally manipulated calls. We found that GPs maintained their call categorization across a wide range of gross temporal manipulations such as changes to tempo and altered ISI distributions. In contrast, GP behavior was strongly affected by altering the F0 of calls. In parallel, to determine which features GPs might be using to extract knowledge of call category from stimuli, we extended a previously developed feature-based model of auditory categorization by adding a WTA feature-integration stage that enabled us to obtain a categorical decision from the model on a trial-by-trial basis. We trained the model using natural GP calls. When we challenged the model with the identical stimuli used in the GP experiments, we obtained model responses that explained ~50% of the overall variance in GP behavior. This result suggests that learning to generalize over the variability in natural call categories resulted in the model being able to generalize over some artificial call manipulations. That the model tracks GP performance on these manipulated stimuli is a key observation that supports our reasoning that GPs learn model-like features to generalize over call variability and accomplish call categorization. We had previously reported electrophysiological support for the feature-based model by demonstrating that a large fraction of neurons in the superficial layers of auditory cortex exhibited feature-selective responses, resembling the FD stage of the model (*Montes-Lourido et al., 2021a*). The results described in the present manuscript lend further support to the model at a behavioral level. Taken together, these studies strongly suggest how a spectral content-based representation of sounds at lower levels of auditory processing may be transformed into a goal-directed representation at higher processing stages by extracting and integrating task-relevant features.

The feature-based model was highly predictive of GP behavior, although it was conceptualized from purely theoretical considerations, trained only using natural GP calls, and implemented without access to any behavioral data. We developed the feature-based model to capture how knowledge of auditory categories is learned by encoding informative features, but did not account for how this knowledge is expressed during behavior. Rather, we made minimal assumptions and used a WTA framework with a static decision criterion (with a small amount of error). Insights from our behavioral observations could

be used to further refine the model by adding a realistic behavioral expression stage. For example, our data indicated that GPs altered their behavioral strategy over the course of multiple sessions within a given day. This could possibly reflect an early impulsivity in their decision-making brought on by food deprivation (evidenced by a FA rate) that gradually switches to a punishment-avoidance strategy with increasing satiation (although $d'$ remains consistent across sessions). Additional arousal-dependent effects on behavior are likely. Nevertheless, the fact that the current model could explain ~50% of the variance in behavioral trends we observed suggests that the fundamental strategy employed by the model—that of detecting features of intermediate complexity to generalize over within-category variability—also lies at the core of GP behavior.

Furthermore, we could leverage the model to gain insight into possible behavioral strategies used by GPs in performing the tasks. For example, we could compare models trained to categorize calls in one versus many or one versus one conditions to ask which scenario was more consistent with GP behavior: (1) whether the GPs used prior features that they acquired over their lifetimes to categorize a given call type from all other calls, or (2) whether GPs were de-novo learning new features to solve the particular categorization task on which they were trained. While the relatively lower volume of data used to train the one vs. one model is a potential confound, the model trained on call features that distinguish a particular call from all other calls was more closely aligned with GP behavioral data, supporting the first possibility. Examining how different subsets of features could be employed to solve different categorization tasks revealed possible strategies that GPs might use to flexibly recruit different feature representations to solve our tasks. While we have used GPs as an animal model for call categorization in this study, we have previously shown that the feature-based model shows high performance across species (GPs, marmosets, and macaques), and that neurons in the primary auditory cortex (A1) of marmosets also exhibit feature-selective responses (*Liu et al., 2019*). Thus, it is likely that our model reflects general principles that are applicable across species, and offers a powerful new approach to deconstruct complex auditory behaviors.

On the behavioral side, previous GP studies have largely used conditioned avoidance (e.g., *Heffner et al., 1971*) or classical conditioning using aversive stimuli (e.g., *Edeline et al., 1993*) to study simple tone detection and discrimination behaviors. Studies probing GP behaviors using more complex stimuli are rare (e.g., *Ojima et al., 2012*; *Ojima and Horikawa, 2015*). Our study of GP call categorization behavior using multiple spectrotemporally rich call types and parametric manipulations of spectral and temporal features offers comprehensive insight into cues that are critical for call categorization and builds significantly on previous studies. First, we showed that GPs can categorize calls in challenging SNRs, and that thresholds vary depending on the call types to be categorized. We demonstrated that information for GP call categorization was available in short-duration segments of calls, and consistent with some previous studies in other species (*Holfoth et al., 2014*; *Knudsen and Gentner, 2010*; *Marslen-Wilson and Zwitserlood, 1989*; *Pitcher et al., 2012*), GPs could extract call category information soon after call onset. GP call perception was robust to large temporal manipulations, such as reversal and larger changes to tempo than have been previously tested (*Neilans et al., 2014*). These results are also consistent with the resilience of human word identification to large tempo shifts (*Janse et al., 2003*). Our finding that GP call categorization performance is robust to ISI manipulations is also not necessarily inconsistent with results from mice (*Perrodin et al., 2020*); in that study, while female mice were found to strongly prefer natural calls compared to calls with ISI manipulations, it is possible that mice still identified the call category correctly. For gross spectral manipulations, we found that GP call categorization was robust to a larger range of F0-shifts than have been previously tested (*Neilans et al., 2014*). Critically, for all but one of these manipulations, the feature-based model captured GP behavioral trends with surprising accuracy both qualitatively and quantitatively.

An analysis of model deviation from behavior could suggest a roadmap for future improvements to our model that could yield further insight into auditory categorization. The one paradigm where we observed a systematic under-performance of the model compared to GP behavior was when we presented call segments of varying lengths from call onset. While the GPs were able to accomplish categorization by extracting information from as little as 75 ms segments, the model required considerably more information (~150 ms). This is likely because the model was based on the detection of informative features that were on average of ~110 ms duration, which were identified from an initial random set of features that could be up to 200 ms in duration. We set this initial limit based on

an upper limit estimated from electrophysiological data recorded from A1 (*Montes-Lourido et al., 2021a*). We consciously did not impose further restrictions on feature duration or bandwidth to ensure that the model did not make any assumptions based on observed behavior. Upon observing this deviation from behavior, to test whether restricting feature lengths to shorter durations would further improve model performance, we repeated the modeling by constraining the initial feature length to a maximum of 75 ms (the lowest duration for which GPs show above-threshold performance). We found that the constrained MIF model better matched GP behavior on the segment-length task ($R^2$ of 0.62 and 0.58 for the chuts vs. purrs and wheeks vs. whines tasks) with the model exceeding $d'$=1 for 75 ms segments for most tested cases. The constrained MIF model maintained similarity to behavior for the other manipulations as well (*Figure 11—figure supplement 2*), and yielded higher overall $R^2$ values (0.66 for chuts vs. purrs, 0.51 for wheeks vs. whines), explaining ~59% of the variance in GP behavior. This result illustrates how behavioral experiments can provide constraints for the further development of theoretical models in the future.

We also observed the over-performance of the model compared to behavior in some paradigms. Some of this over-performance might be explained by the fact that the model does not exhibit motivation changes, and so forth, as outlined above. A second source of this over-performance might arise from the fact that our model integrates evidence from the FD stage perfectly, that is, we take the total evidence for the presence of a call category to be the weighted sum of the log-likelihoods of all detected features (counting detected features only once) over the entire stimulus, and do not explicitly model a leaky integration of feature evidence over time, as is the case in evidence-accumulation-to-threshold models (*Cheadle et al., 2014*; *Keung et al., 2020*). Future improvements to the model could include a realistic feature-integration stage, where evidence for a call category is generated when a feature is detected and degrades with a biologically realistic time constant. In this case, a decision threshold could be reached before the entire stimulus is heard, but model parameters would need to be derived from or fit to observed behavioral data (*Glaze et al., 2015*).

How do the proposed model stages map onto the auditory system? In an earlier study, we provided evidence that feature detection likely occurs in the superficial layers of A1, in that a large fraction of neurons in this stage exhibits highly selective call responses and complex spectrotemporal receptive fields (*Montes-Lourido et al., 2021a*). How and at what stage these features are combined to encode a call category remains an open question. Neurons in A1 can acquire categorical or task-relevant responses to simple categories, for example, low versus high tone frequencies, or low versus high temporal modulation rates, with training (*Bao et al., 2004*; *Fritz et al., 2005*). In contrast, categorical responses to more complex sounds or non-compact categories only seem to arise at the level of secondary or higher cortical areas or the prefrontal cortex (*Russ et al., 2008*; *Yin et al., 2020*), which may then modulate A1 via descending connections. These results, taken together with studies that demonstrate enhanced decodability of call identity from the activity of neurons in higher cortical areas (*Fukushima et al., 2014*; *Grimsley et al., 2012*; *Grimsley et al., 2011*), suggest that secondary ventral-stream cortical areas, such as the ventral-rostral belt in GPs, are promising candidates as the site of evidence integration from call features. The WTA stage may be implemented via lateral inhibition at the same level using similar mechanisms as has been suggested in the primary visual cortex (*Chettih and Harvey, 2019*) or may require a further upstream layer. Further experiments are necessary to explore these questions.

The feature-based model we developed offers a trade-off between performance and biological interpretability. Modern deep neural network (DNN) based models can attain human-level performance (e.g., in vision: *Rajalingham et al., 2015*, in audition: *Kell et al., 2018*) but what features are encoded at the intermediate network layers remain somewhat hard to interpret. These models also typically require vast quantities of training data. In contrast, our model is based on an earlier model for visual categorization (*Ullman et al., 2002*) that is specifically trained to detect characteristic features that contrast the members of a category from non-members. Thus, we can develop biological interpretations for what features are preferably encoded and more importantly, why certain features are more advantageous to encode. Because the features used in the model are the most informative parts of the calls themselves, they can be identified without a parametric search. This approach is especially well-suited for natural sounds such as calls that are high-dimensional and difficult to parameterize. We are restricted, however, in that we do not know all possible categorization problems that are relevant to the animal. By choosing well-defined categorization tasks that are ethologically critical for

an animal's natural behavior (such as call categorization in the present study), we can maximize the insight that we can derive from these experiments as it pertains to a range of natural behaviors. In the visual domain, the concept of feature-based object recognition has yielded insight into how human visual recognition differs from modern machine vision algorithms (*Ullman et al., 2016*). Our results lay the foundation for pursuing an analogous approach for understanding auditory recognition in animals and humans.

## Materials and methods

All experimental procedures conformed to the NIH Guide for the use and care of laboratory animals and were approved by the Institutional Animal Care and Use Committee of the University of Pittsburgh (protocol number 21069431).

### Animals

We acquired data from four male and three female adult, wild-type, pigmented GPs (Elm Hill Labs, Chelmsford, MA), weighing ~500–1000 g over the course of the experiments. After a minimum of 2 weeks of acclimatization to handling, animals were placed on a restricted diet for the period of behavioral experiments. During this period, GPs performed auditory tasks for food pellet rewards (TestDiet, St. Louis, MO). The weight and body condition of animals were closely monitored and the weight was not allowed to drop below 90% of baseline weight. To maintain this weight, depending on daily behavioral performance, we supplemented GPs with restricted amounts of pellets (~10–30 g), fresh produce (~10–30 g), and hay (~10–30 g) in their home cages. All animals had free access to water. After behavioral testing for ~2–3 weeks, animals were provided ad-libitum food for 2–3 days to obtain an updated estimate of their baseline weights.

### Behavioral setup

All behavioral tasks were performed inside a custom behavioral booth (*Figure 1*; ~90×60×60 cm³) lined with ~1.5 cm thick sound attenuating foam (Pinta Acoustic, Minneapolis, MN) (*Figure 1A*). The booth was divided into two halves (~45×60×60 cm³ each) using transparent acrylic (McMaster-Carr, Los Angeles, CA). One half contained the behavioral setup. The other half was sometimes used as an observation chamber in which we placed a naive GP to observe an expert GP perform tasks; such social learning has been shown to speed up behavioral task acquisition (*Paraouty et al., 2020*). The entire booth was uniformly lit with LED lighting. The behavioral chamber contained a 'home base' area and a reward region (*Figure 1B*). A water bottle was placed in the home base to motivate animals to stay at/return to the home base after each trial. A pellet dispenser (ENV-203-45, Med Associates, Fairfax, VT) was used to deliver food pellets (TestDiet) onto a food receptacle placed in a corner of the reward area. Air puffs were delivered from a pipette tip placed near the food receptacle directed at the animal's snout. The pipette tip was connected using silicone tubing via a pinch valve (EW98302-02, Cole-Palmer Instrument Co., Vernon Hills, IL) to a regulated air cylinder, with the air pressure regulated to be about 25 psi.

The animal's position within the behavioral chamber was tracked using MATLAB (Mathworks, Inc, Natick, MA) at a video sampling rate of ~25 fps using a web camera (Lifecam HD-3000, Logitech, Newark, CA) placed on the ceiling of the chamber. Sound was played from a speaker (Z50, Logitech) located ~40 cm above the animal at ~70 dB SPL with a sampling frequency of 48 kHz. Pellet-delivery, illumination, and air puff hardware were controlled using a digital input/output device (USB-6501, National Instruments, Austin, TX).

### Basic task design

All behavioral paradigms were structured as Go/ No-go tasks. GPs were required to wait in the home base (*Figure 1B*) for 3–5 s to initiate a trial. A Go or No-go stimulus was presented upon trial initiation. For Go stimuli, moving to the reward area (*Figure 1B*) was scored as a hit and resulted in a food pellet reward; failure to do so was scored as a miss. For No-go stimuli, moving to the reward area was scored as a FA and was followed by a mild air puff and brief time-out with the lights turned off (*Figure 1A*), whereas staying in the home base was scored as a correct rejection.

## Training GPs via social learning and appetitive reinforcement

Naïve animals were initially placed in the observer chamber while an expert GP performed the task in the active chamber. Such social learning helped accelerate forming an association between sound presentation and food reward (*Paraouty et al., 2020*). Following an observation period of 2–3 days, naive GPs were placed in the active chamber alone and underwent a period of Pavlovian conditioning, where Go stimuli were played, and food pellets were immediately dropped until the animals built an association between the sound and the food reward. Once GPs began to reliably respond to Go stimuli, No-go stimuli along with the air puff and light-out were introduced at a gradually increasing frequency (until about equal frequency of both Go and No-go stimuli). We trained two cohorts of four adult GPs ( two males and two females) for two call categorization tasks (as discussed later), with the overlap of one GP between the tasks.

## Stimuli and behavioral paradigms

### Learning

In this study, we trained GPs to categorize two similar low-frequency, temporally symmetric, affiliative call categories ('*chuts*'—Go and '*purrs*'—No-go, *Figure 1C*), or two temporally asymmetric call categories with non-overlapping frequency content ('*wheeks*'—Go and '*whines*'—No-go, *Figure 1D*). All calls were recorded in our laboratory as described earlier (*Montes-Lourido et al., 2021b*) and were from animals unfamiliar to the GPs in the present study. Calls were trimmed to ~1 s length, normalized by their rms amplitudes, and presented at ~70 dB SPL (*Figure 1C, D*). Different sets of randomly selected calls, each set containing eight different exemplars, were used for the learning and generalization phases. Other paradigm-specific stimuli were generated by manipulating the call sets used during the learning phase as explained below. We first manually trained animals to associate one corner of the behavioral chamber with food pellet rewards. Following manual training, we began a conditioning phase where we only presented Go stimuli when the animal was in the home base area followed by automated pellet delivery, gradually increasing the interval between stimulus and reward. Once animals began moving toward the reward location in anticipation of the reward, we gradually introduced an increasing proportion of No-go stimuli, and began tracking the performance of the animal. During the learning phase, animals typically performed the Go/No-go task for 6 sessions each day with ~40 trials per session. Each session typically lasted ~10 min.

### Generalization to new exemplars

Once animals achieved $d'>1.5$ on the training stimulus set, we replaced all training stimuli with eight new exemplars of each call category that the animals had not heard before. To minimize exposure to the new exemplars, we tested generalization for about 3 days per animal, with 1–2 sessions with training exemplars and 1–2 sessions of new exemplars.

### Call-in-noise

To generate call-in-noise stimuli at different SNRs, we added white noise of equal length to the calls (gated noise) such that the SNR ratio, computed using rms amplitudes, varied between –18 dB and +12 dB SNR (i.e., –18, –12, –6, 3, 0, +3, +6, and +12 dB SNR). This range of SNRs was chosen to maximize sampling over the steeply growing part of psychometric curve fits. We presented these stimuli in a block design, measuring GP behavior in sessions of ~40 trials with each session having a unique SNR value. We typically collected data for three sessions for each of the nine SNR levels including the clean call. SNR data were collected across several days per animal, with different SNRs tested each day to account for possible fluctuations in motivation levels.

### Restricted segments

To investigate how much information is essential for GPs to successfully categorize calls, we created call segments of different lengths (50, 75, 100, 125, 150, 175, 200, 300, 400, 500, 600 700, and 800 ms) from the call onsets. We chose 800 ms as the maximum segment length since our briefest call was ~800 ms long. We tested GPs on these 13 segment lengths, presenting 5 repetitions of 8 exemplars per category. A randomized list of all 1040 trials was created (2 categories×8 exemplars×13 time-chunk

lengths×5 repetitions) and presented sequentially in sessions of ~40 trials, completing ~240 trials per day (~5 days to complete the entire list of stimuli).

### Tempo manipulation

To temporally compress or stretch the calls without introducing any alterations to long-term spectra, we changed the tempo of the calls using Audacity software (high-quality option, using a Subband Sinusoidal Modeling Synthesis algorithm). The algorithm detects peaks in the short-term Fourier Transform of short sound segments and resynthesizes the segment using a phase-preserving oscillator bank at the required time scale. Tempo was changed by –120%, –100%, –80%, –60%, –30%, +30%, +60%, and +80% which resulted in calls that were ~0.45, 0.5, ~0.56, ~0.63, ~0.77, ~1.43, 2.5, and 5 times the original lengths of the calls, respectively. As earlier, 720 total trials were presented (2 categories×8 exemplars×9 tempo conditions×5 repetitions).

### ISI manipulations

To determine if GPs used individual syllables or temporal patterns of syllables for call categorization, we introduced several ISI manipulations, while keeping the individual syllables intact. After manually identifying the beginnings and endings of each syllable within the calls, the syllables and the ISI values were extracted using MATLAB. Since our recorded calls have some level of background noise, we first created a set of control stimuli where the audio in the ISI was replaced with silence. As a second control, we changed the ISI values by randomly drawing ISI values from the ISI distribution of the same call category. Five such new calls were generated from each original call. We acquired behavioral responses using a randomized presentation strategy as above, split into (1) 640 trials with regular ISI (with background recording noise) and silent ISI (2 categories×8 exemplars×2 conditions×20 repetitions), and (2) 400 trials with random within-category ISI values (2 categories×8 exemplars×5 random ISI combinations×5 repetitions). We then generated chimeric calls with syllables belonging to one category and ISI values belonging to the other category (e.g., chut syllables with purr ISI values). Five such chimeric calls were created per original call. Because these calls contain information from both categories, we adopted a catch-trial design for this experiment. Natural calls (Syllable and ISI from the same category, both Go and No-go categories) were presented on 67% of trials, and chimeric calls on 33% of trials (catch trials). We rewarded 50% of the catch trials at random and did not provide any negative reinforcement (air puff or time-out). Thus, 1200 randomized trials were presented, with 800 trials with regular calls and 400 catch trials with chimeric calls.

### Call reversal

As a final gross temporal manipulation, we temporally reversed the calls. We presented a total of 160 trials in randomized order (2 categories×8 exemplars×2 conditions×5 repetitions) for this experiment.

### Fundamental frequency manipulation

We created calls with fundamental frequency (F0) varying from one octave lower and to one octave higher by changing the pitch of the calls by –50%, –40%, –30%, –20%, 20%, 40%–50%, and 100% using Audacity software. These pitch changes re-interpolated the calls such that call length and tempo were preserved. A total of 720 trials (2 categories×8 exemplars×9 F0 conditions×5 repetitions) were presented in randomized order for this experiment.

### Low pass filtering

For the wheeks versus whines task, we low pass filtered both wheeks and whines at 3 kHz using a 256-point FIR filter in MATLAB. We presented 160 trials (2 categories×8 exemplars×2 conditions×5 repetitions) in randomized order for this experiment.

## Analysis of behavioral data

All analysis was performed in MATLAB. Specific functions and toolboxes used are mentioned where applicable below.

To quantify the behavioral performance of the animals, we used the sensitivity index or $d'$ (***Green and Swets, 1966***), defined as:

$$d' = Z(H) - Z(FA) \tag{1}$$

where $H$ and $FA$ represent the hit rate and FA rate, respectively. To avoid values that approach infinity, we imposed a floor (0.01) and ceiling (0.99) on hit rates and FA rates.

For the learning and generalization data, the $d'$ value was estimated per session using the H and FA rates from that session. These session-wise hit rates, FA rates and $d'$ estimates were averaged for each animal and the mean and standard error of mean (s.e.m.) across all animals are reported in the results section.

For all the call manipulation experiments (including call-in-noise), a single hit rate, FA rate and $d'$ were estimated per condition per animal by pooling data over all trials corresponding to each condition. The mean and SEM of these indices across all animals are reported in the results section.

Additionally, for the call-in-noise data, we used the 'fitnlm' MATLAB function (Statistics toolbox) to fit psychometric functions of the form (*Wichmann and Hill, 2001*):

$$\psi(x; \alpha, \beta, \lambda) = (1 - \lambda) * F(x; \alpha, \beta) \tag{2}$$

where $F$ is the Weibull function, defined as $F(x; \alpha, \beta) = 1 - exp\left(-\left(\frac{x}{\alpha}\right)^{\beta}\right)$, $\alpha$ is the shift parameter, $\beta$ is the slope parameter, and $\lambda$ is the lapse rate.

## Statistical analyses

We used paired t-tests to compare $d'$ values across animals in experiments with only two conditions, that is, reversal and low-pass filtering. For the remaining experiments with more than two conditions (segment length, SNR, frequency shift, and tempo shift), we fit generalized linear mixed-effect models with logit link functions to the binary (Go/Nogo) behavioral and model responses on a trial-by-trial basis using the stimulus type (Go/Nogo), parameter value (for example, SNR level or frequency shift magnitude), and for length-shift and frequency-shift experiments where positive as well as negative shifts were possible, parameter sign as predictors, an interaction term between stimulus type and parameter value, and animal ID as a random effect (*Equations 3 and 4*). To evaluate whether the parameter value and sign significantly modulated behavioral responses after accounting for stimulus type, we compared the above full models to null models that had only the stimulus type as a predictor and animal ID as a random effect (*Equation 5*). We fit these models in R (version 4.2.1) using the 'glmer' function in the lme4 package (*Bates et al., 2014*), and compared the full and null models using the ANOVA function in the 'stats' package in R. Statistical outputs from R are available as supplementary information (*Supplementary file 1*).

Full model (SNR and segment length manipulations):

$$Response \sim StimulusType + ParameterValue + StimulusType : ParameterValue + (1 \mid AnimalID) \tag{3}$$

Full model (frequency shift and length shift manipulations):

$$Response \sim StimulusType + ParameterValue + ParameterSign + StimulusType : ParameterValue + (1 \mid AnimalID) \tag{4}$$

Null model:

$$Response \sim StimulusType + (1 \mid AnimalID) \tag{5}$$

Finally, for the swapped ISI stimuli in the ISI manipulation experiments, since we did not have well-defined categories for the chimeric calls, we chose to compare the Go-rates for the stimuli with syllables of one kind using a paired *t*-test. All statistical outputs are available in supplementary material (*Supplementary file 2*).

## Feature-based categorization model

To gain insight into what potential spectrotemporal features GPs may be using to accomplish call categorization in the behavioral tasks, we extended a previously published feature-based model that achieves high classification performance for categorizing several call types across several species, including GPs (*Liu et al., 2019*). The model consists of three layers: (1) a spectrotemporal

representational layer, (2) a feature detection (FD) layer, and (3) a competitive WTA decision layer. The first two layers are closely based on *Liu et al., 2019*; we briefly describe these stages below. The WTA layer combines information from the FD layer to form a Go/No-go decision.

The spectrotemporal representational layer consisted of the output of a biologically realistic model of the auditory periphery (*Zilany et al., 2014*). Cochleagrams of training and testing calls were constructed from the inner-hair-cell voltage output of this model (*Zilany et al., 2014*). Cochleagrams were constructed using 67 characteristic frequencies logarithmically spaced between 200 Hz and 20 kHz and were sampled at 1 kHz. Model parameters were set to follow healthy inner and outer hair cell properties and cat frequency tuning.

For the FD layer, we trained four separate sets of feature detectors to classify the four call types, where each set classified a single call type (e.g., chut) from all other call types (i.e., a mixture of purr, wheek, whine, and other calls). During training, for each call type, we identified a set of maximally informative features (MIFs; see *Liu et al., 2019*, based on an algorithm developed by *Ullman et al., 2002*) that yielded optimal performance in classifying the target call type from other call types (*Figure 2*). To do so, we generated an initial set of 1500 candidate features by randomly sampling rectangular spectrotemporal blocks from the target call cochleagrams. We restricted the duration of features to a maximum of 200 ms, based on typically observed temporal extents of spectrotemporal receptive fields in superficial layers of the GP primary auditory cortex (*Montes-Lourido et al., 2021a*). Next, we evaluated how well each feature classified the target call type from other call types. To do so, we obtained the maximum normalized cross-correlation value ($r_{max}$) of each feature with target calls and other calls. Each feature was assigned a threshold that indicated if the feature was detected in the stimulus ($r_{max}$ >threshold) or not ($r_{max}$ < threshold). We used mutual information to determine the utility of each feature in accomplishing the classification task. By testing a range of threshold values, we obtained the optimal threshold for each feature at which its categorization was maximal. The log-likelihood ratio of this binary classification was taken to be the weight of each feature. From this initial random set of 1500 features, we used a greedy search algorithm to obtain the set of maximally informative and least redundant features that achieved optimal performance to classify the training data set. The maximum number of these features was constrained to 20. The training performance of the MIF set was assessed by first estimating the receiver operating characteristic curve and then quantifying the area under the curve (AUC), using the procedure described in *Liu et al., 2019*. To ensure the robustness of these solutions, we generated five instantiations of the MIFs for classifying each call type by iteratively determining an MIF set and removing these features from the initial set of features when training the next MIF set. We verified that training performance did not drop for any of these five instantiations.

Next, to compare model performance with GP behavioral performance, we evaluated model performance in classifying the same stimuli used in the behavioral experiments using the sensitivity metric, *d'*. To simulate the Go/No-go task, we employed a WTA framework, as described below. In a single trial, the stimulus could either be a target (Go stimulus) or a distractor (No-go stimulus). For this stimulus, we estimated the target FD-layer response as the sum of detected (target) MIF weights normalized by the sum of all (target) MIF weights. This normalization scales the model response to a range between 0 (no MIFs detected) and 1 (all MIFs detected). Similarly, we estimated the distractor model response as the sum of detected (distractor) MIF weights normalized by the sum of all (distractor) MIF weights. If the target FD-stage response was greater (less) than the distractor FD-stage response, then the WTA model would predict that the stimulus in that trial was a target (distractor). To allow for non-zero guess rate and lapse rate, as typically observed in behavioral data, we set the minimum and maximum Go probability of the WTA output to 0.1 and 0.9 (*Figure 2C*). These Go probabilities [$P_{trial-n}(GO)$] were realized on a trial-by-trial basis where a random number ($X$) drawn from a uniform distribution between 0 and 1 was compared with the WTA model Go probability to decide the final response [Go if $X < P_{trial-n}(GO)$]. *d'* was estimated from the hit rate and false alarm rate using *Equation 1*. Identical test stimuli and number of trials were used for both behavior and model. We treated each of the five instantiations of the MIFs as a unique 'subject' for analysis.

## Spectrum-based categorization model

To determine whether call categories could be discriminated based on long-term spectral content, we constructed a SVM classifier. The long-term power spectrum (estimated from the cochleagram as the

variance of the mean rate at each center frequency) was used as input to the model. Similar to feature-based models, SVMs were trained to classify a single call type from all other call types using the Matlab function *fitcsvm* (linear kernel, standardized [i.e., rescaled] predictors). The tenfold cross-validation losses for chut, purr, wheek, and whine SVMs were 14%, 4%, 10.6%, and 7.1%, which indicate small bias/variance for the SVM models. Similar to the feature-based model, a WTA stage was implemented by comparing the outputs of the target-call SVM model and the distractor-call SVM model for a single input call (response=GO if target-SVM output >distractor SVM output). We then used this model to classify the manipulated stimuli and derive $d'$ values for comparison with GP behavior.

## Acknowledgements

This work was supported by the National Institutes of Health, NIH R01DC017141 (SS) and start-up funds from the University of Pittsburgh. Dr Shi Tong Liu developed MATLAB code implementing the feature-based auditory categorization model. The authors are grateful to Dr Bryan Pfingst and Deborah Colesa (University of Michigan) for detailed advice regarding the food pellet rewards and pellet dispenser. Samuel Li assisted with the recording and curation of GP calls used in this study. The authors thank Jillian Harr, Sarah Gray, Julia Skrinjar, Brent Barbe, and Elizabeth Chasky for assistance with animal care. This research was supported in part by the University of Pittsburgh Center for Research Computing through the resources provided.

## Additional information

### Funding

| Funder | Grant reference number | Author |
| --- | --- | --- |
| National Institutes of Health | R01DC017141 | Srivatsun Sadagopan |
| University of Pittsburgh | | Srivatsun Sadagopan |

The funders had no role in study design, data collection and interpretation, or the decision to submit the work for publication.

### Author contributions

Manaswini Kar, Conceptualization, Data curation, Formal analysis, Investigation, Visualization, Methodology, Writing – review and editing; Marianny Pernia, Investigation, Methodology, Writing – review and editing; Kayla Williams, Investigation, Writing – review and editing; Satyabrata Parida, Data curation, Software, Formal analysis, Investigation, Visualization, Methodology, Writing – original draft, Writing – review and editing; Nathan Alan Schneider, Software, Investigation, Methodology; Madelyn McAndrew, Isha Kumbam, Investigation; Srivatsun Sadagopan, Conceptualization, Resources, Software, Formal analysis, Supervision, Funding acquisition, Validation, Methodology, Writing – original draft, Project administration, Writing – review and editing

### Author ORCIDs

Marianny Pernia http://orcid.org/0000-0002-9889-3577
Satyabrata Parida http://orcid.org/0000-0002-2896-2522
Nathan Alan Schneider http://orcid.org/0000-0002-9145-5427
Srivatsun Sadagopan http://orcid.org/0000-0002-1116-8728

### Ethics

All experimental procedures conformed to the NIH Guide for the Care and Use of Laboratory Animals and were approved by the institutional animal care and use committee of the University of Pittsburgh (protocol number 21069431).

### Decision letter and Author response

Decision letter https://doi.org/10.7554/eLife.78278.sa1
Author response https://doi.org/10.7554/eLife.78278.sa2

## Additional files

### Supplementary files

- Supplementary file 1. Statistical outputs of GLM model comparisons.
- Supplementary file 2. Statistical outputs of other tests used in the manuscript.
- Transparent reporting form

### Data availability

All data generated or analyzed during this study are included in the manuscript and supporting file; Source Data files have been provided for Figures 3 - 12.

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
