## [Editor Report]

This important study combines behavioral data from guinea pigs and data from a classifier model to make a compelling case for which auditory features are important for classifying vocalisations. This study is likely to be of interest to both computational and experimental neuroscientists, in particular auditory neurophysiologists and cognitive and comparative neuroscientists. A strength of this work is that a model trained on natural calls was able to predict some aspects of responses to temporally and spectrally altered cues.

---

## [Decision Letter]

**Decision letter after peer review:**

Thank you for submitting your article "Vocalization categorization behavior explained by a feature-based auditory categorization model" for consideration by *eLife*. Your article has been reviewed by 3 peer reviewers, including Dan FM Goodman as Reviewing Editor and Reviewer #1, and the evaluation has been overseen by Andrew King as the Senior Editor. The following individual involved in the review of your submission has agreed to reveal their identity: Patrick Krauss (Reviewer #3).

Essential revisions:

1. The statistical analysis seems to be underpowered. Reviewer #2 makes a suggestion for another approach.

2. The data and models do not seem to be sufficient to support some of the stronger claims made in the manuscript. Please revise the language.

*Reviewer #1 (Recommendations for the authors):*

I think this is a really nice paper. I love the approach and personally find the conclusions interesting and compelling. I think maybe the case for the strength of the predictions is somewhat overstated and the manuscript would be improved by toning that down a bit and acknowledging rather than explaining away divergences between model and data.

In figure 4, the R^2^ values are both 0.939. Can you confirm that this isn't an error? It seems surprising that they should have this identical value. Might be good to double-check that.

In figure 6, the model seems to perform best for a time stretch value not equal to 1. This seems surprising. Any thoughts on that?

In figure 7, what is being tested for the model? ISI drawn from same-category or other-category (chimeric)? Maybe show both separately? Generally, it's a bit hard to work out what's going on in this figure.

In figure 8C the decrease in performance of the model is referred to in the text as "only a slight decrease in d'" but this doesn't seem very consistent with what looks like quite a large decrease in the figure.

In the methods, for the tempo manipulation, could you explain more what the method is? I guess it is some sort of pitch-preserving transformation that is built into Audacity, but without knowing how that software works it's hard to know what's going on here.

*Reviewer #2 (Recommendations for the authors):*

I have concerns about the statistical analysis of the behavioral data and the main conclusions drawn from the study. A major weakness in the current manuscript is the statistical approach used for analyzing the behavior which is in all cases likely to be underpowered – given that the absence of an effect (e.g. pitch roving) is often taken as a key result, a more rigorous approach is needed (see below for detailed points). This could either take the form of Bayesian statistics which the likelihood of a significant effect or a null effect can be distinguished from an inconclusive effect (my guess is most tests run on 3 animals will be inconclusive) or better a mixed effect model in which the binary trial outcome is modelled according to e.g. the call type, pitch shift/temporal modulation with the individual GP as a fixed effect. This is likely to be a more sensitive analysis and the resulting β coefficients may be informative and may enable a more direct comparison to the model output (if the model is run on single trials, like the GP, then the exact same analysis could be performed on different model instantiations). It would also be possible to combine all animals into the same analysis with the task as a factor.

In terms of the interpretation; the classifier is essentially only looking at the spectrum (few of the statistics that e.g. Josh McDermott's group has identified as critical components of natural sounds are captured, such as frequency or amplitude modulation). This approach works for these highly reduced two-choice discriminations that can essentially be solved by looking at spectral properties, although the failures of the model to cope with temporal manipulations suggest its overfitting to the trained exemplars by looking at particular temporal windows. From the spectrograms it is clear that the temporal properties of the call are likely to be important – e.g. the purr is highly regular; a property that is maintained across different presentation rates, likely explaining why the GPs cope fine with short rates. Many of the findings here replicate Ojima and Horikawa (2016) – this should be stated explicitly within the manuscript. The behavioral data here are impressive but I'm not really sure what we should take away from the manuscript beyond that, except that for an unnaturally simple task an overly simple frequency-based classifier can solve the task in some instances and not others.

Detailed comments

Figure 3 – what is the performance of the trained exemplars here? From the other figures I guess a lot higher than for the generalization stimuli; it looks like performance does take a hit on novel exemplars (possibly for GPs and the model?). The authors should justify why they didn't present stimuli as rare (and randomly rewarded) probe stimuli to assess innate rather than learned generalization.

Figure 5. – segment length analysis – here the GP is much better at short segments. The model and the GP show a clear modulation in performance with segment length that is not picked up statistically due to an underpowered analysis. These data should either be analysed as a within animal logistic regression or an across animal mixed effect model. This could combine all 6 animals, with the task as a factor.

Figure 6 – again, repeated measures anova on 7 conditions in 3 animals (χ2); the fact that the p-value is >0.05 means nothing! The real question – is the model quantitatively similar (could be addressed by assessing the likelihood of getting the same performance by generating a distribution of model performance on many model instantiations and then asking if the GP data falls within 95% of the observed values) or is it qualitatively similar? – for example show the same variation in performance across different call rates. In F the GP performance is robust across all durations, with a gradual increase in performance from the shortest call durations. In contrast the model peaks around the trained duration. In E both the model and the GP peak at just less than 1x call duration, but the model does really poorly at short durations. To me, this isn't 'qualitatively similar'.

Figure 7 – why use the term chimeric in this figure and nowhere else? Are they not just the random ISI as in the second half of the figure?

Figure 8 – I'm surprised at how poorly the model does on the reversed chuts. Again, the lack of a significant difference here cannot be used as evidence for equivalent performance.

Figure 12 – I think I possibly don't understand this figure – how can the MIFs well suited for a purr vs chut distinction be different from a chut vs purr distinction? Either this needs elaboration or I think It would make far more sense to display chuts and whines (currently supplemental figure 12). Moreover, rather than these rather unintuitive bubble plots, a ranked list of each feature's contribution to the model, partial dependence plots, and individual conditional expectation plots would answer the question of which model features most contribute towards a guinea pig call categorization.

The model classifies the result based on weighted factors such as spectrotemporal information but does not consider specific biological neuronal networks that underly this categorization process. Moreover, the authors assert that intermediate-complexity features were responsible for call categorization; this assertion is expected as all MIFs selected are based on the inputted spectrotemporal data. In other words, the features in the classifier model could arguably said to be mid-complex as all vocalizations could be said to have inherent complexity. Together, the conclusions that the authors draw that this model could represent a biological basis for call categorization seem really to reach beyond the data; the model fits some aspects of the training set well but does not causally prove that GPs learn specific frequencies to represent calls early in their lives.

*Reviewer #3 (Recommendations for the authors):*

This is a great study! I appreciate the approach to complementing behavioral experiments with computational modelling.

Considering the temporal stretching/compressing and the pitch change experiment: it remains unclear if the MIF kernels used by the models were just stretched/compressed to compensate for the changed auditory input. If so, the modelling results are kind of trivial. In this case, the model provides no mechanistic explanation of the underlying neural processes. This issue should at least be addressed in the corresponding Results section and might be an interesting cornerstone for a follow-up study.

---

## [Author Response]

Essential revisions:1. The statistical analysis seems to be underpowered. Reviewer #2 makes a suggestion for another approach.

We have implemented the alternate statistical analyses suggested by reviewer 2 in this revision. These analyses involve using a generalized linear model to fit the behavioral data on a trial-by-trial basis. The alternate statistical analyses do not substantially alter the fundamental conclusions of the manuscript, i.e., the comparison of models to behavioral data is unaltered.

2. The data and models do not seem to be sufficient to support some of the stronger claims made in the manuscript. Please revise the language.

We have addressed this concern in the following ways. (1) At the beginning of the Results section, we clearly state which aspects of categorization behavior the model is intended to capture, and which aspects are left unmodeled. (2) We have eschewed subjective terms and instead state that the model explains ~50% of the behavioral variance (average of overall R^2^ of 0.6 and 0.37 for the chuts vs. purrs and wheeks vs. whine tasks respectively). We leave it to the readers to interpret this effect size.

In addition, we have also addressed all reviewer comments. Please see below for a detailed point-by-point response.

Reviewer #1 (Recommendations for the authors):I think this is a really nice paper. I love the approach and personally find the conclusions interesting and compelling. I think maybe the case for the strength of the predictions is somewhat overstated and the manuscript would be improved by toning that down a bit and acknowledging rather than explaining away divergences between model and data.

Thank you for the constructive feedback! In the revised manuscript we have tried to replace subjective descriptors with actual effect size numbers to avoid overstating results. We also include additional discussion of alternate interpretations.

In figure 4, the R^2^ values are both 0.939. Can you confirm that this isn't an error? It seems surprising that they should have this identical value. Might be good to double-check that.

This is indeed the case for the particular random seed we used to simulate the trial-by-trial model outcome. For other random seeds the values are not equal but in the same ballpark, and therefore we decided to leave these numbers as is in the revised manuscript.

In figure 6, the model seems to perform best for a time stretch value not equal to 1. This seems surprising. Any thoughts on that?

Yes, this is indeed an interesting observation, but while this is the case for the wheeks vs. whines task (Figure 6D), such a trend is not quite clear for the chuts vs. purrs task (Figure 6C). At present we do not have an explanation for why this is the case. Statistically the model did not show a significant effect of tempo change on performance. But interestingly, the Ojima and Horikawa (2016) study observed an asymmetry where GPs maintained task performance for time-compressed stimuli (but not stretched stimuli).

In figure 7, what is being tested for the model? ISI drawn from same-category or other-category (chimeric)? Maybe show both separately? Generally, it's a bit hard to work out what's going on in this figure.

We did not test the model for the chimeric ISI case because we could not assign target and non-target labels to stimuli. This was not clearly mentioned in the original manuscript. Thank you for catching this; we have now added clarification (page 17, line 403).

In figure 8C the decrease in performance of the model is referred to in the text as "only a slight decrease in d'" but this doesn't seem very consistent with what looks like quite a large decrease in the figure.

In figure 8C, the d′ value decreases from 2.76 for regular calls to 2.26 for reversed calls, an 18% decrease. We have replaced “only a slight decrease” with “an ~18% decrease” in the revised manuscript (page 18, line 426).

In the methods, for the tempo manipulation, could you explain more what the method is? I guess it is some sort of pitch-preserving transformation that is built into Audacity, but without knowing how that software works it's hard to know what's going on here.

We used the “high quality” option in the “Change Tempo” feature of audacity. The algorithm used is Subband Sinusoidal Modeling Synthesis (SBSMS), described here: http://sbsms.sourceforge.net/. The fundamental principle is detecting the spectral peaks in the STFT of a small sound segment and resynthesizing the segment with a phase-preserving oscillator bank at the required time scale. This information is included in the Methods section (page 34, line 877).

Reviewer #2 (Recommendations for the authors):I have concerns about the statistical analysis of the behavioral data and the main conclusions drawn from the study. A major weakness in the current manuscript is the statistical approach used for analyzing the behavior which is in all cases likely to be underpowered – given that the absence of an effect (e.g. pitch roving) is often taken as a key result, a more rigorous approach is needed (see below for detailed points). This could either take the form of Bayesian statistics which the likelihood of a significant effect or a null effect can be distinguished from an inconclusive effect (my guess is most tests run on 3 animals will be inconclusive) or better a mixed effect model in which the binary trial outcome is modelled according to e.g. the call type, pitch shift/temporal modulation with the individual GP as a fixed effect. This is likely to be a more sensitive analysis and the resulting β coefficients may be informative and may enable a more direct comparison to the model output (if the model is run on single trials, like the GP, then the exact same analysis could be performed on different model instantiations). It would also be possible to combine all animals into the same analysis with the task as a factor.

We thank the reviewer for this detailed suggestion. As we mention in the response to the reviewer’s public review, we now use a GLM approach to predict trial-wise behavioral and model responses when there are multiple test conditions. Details are in page 36, line 942.

In terms of the interpretation; the classifier is essentially only looking at the spectrum (few of the statistics that e.g. Josh McDermott's group has identified as critical components of natural sounds are captured, such as frequency or amplitude modulation). This approach works for these highly reduced two-choice discriminations that can essentially be solved by looking at spectral properties, although the failures of the model to cope with temporal manipulations suggest its overfitting to the trained exemplars by looking at particular temporal windows. From the spectrograms it is clear that the temporal properties of the call are likely to be important – e.g. the purr is highly regular; a property that is maintained across different presentation rates, likely explaining why the GPs cope fine with short rates.

The classifier is not looking at the average spectrum alone – the reason that the MIFs have a duration of ~110 ms on average is that they are looking for specific spectrotemporal features. In our previous work (Liu et al., 2019), we have shown that constraining the features to small durations (only allowing the model access to instantaneous spectrum information) degrades model performance.

To quantify this further, we developed a simple support vector machine based classifier that attempts to classify vocalization categories based purely on the long-term spectra of calls. To do so, the long-term power spectrum (estimated as the variance of mean rate at each center frequency) was used as input to the model. Similar to feature-based models, SVMs were trained to classify a single call type from all other call types using the Matlab function *fitcsvm* (linear kernel, standardized [i.e., rescaled] predictors). The 10-fold cross-validation losses for chut, purr, wheek, and whine SVMs were 14%, 4%, 10.6%, and 7.1%, which indicate small bias/variance for the SVM models. Similar to the feature-based model, a winner-take-all stage was implemented by comparing the outputs of the target-call SVM model and the distractor-call SVM model for a single input call (response = GO if target-SVM output > distractor-SVM output). We used this model to classify the manipulated stimuli, and found that the spectrum model explained a much lower fraction of the variance in GP behavior.

We have now included this comparison in the revised manuscript (page 21, line 503; Methods in page 39, line 1035) and presented the results of this analysis as Figure 11 —figure supplement 1.

Many of the findings here replicate Ojima and Horikawa (2016) – this should be stated explicitly within the manuscript. The behavioral data here are impressive but I'm not really sure what we should take away from the manuscript beyond that, except that for an unnaturally simple task an overly simple frequency-based classifier can solve the task in some instances and not others.

We respectfully disagree that our findings replicate those of Ojima and Horikawa (2016). That study is indeed impressive in many respects, but the reason that we did not originally cite it is because we chose to focus on vocalizations, and more importantly, on categorization. In this respect, we felt that our stimulus manipulations were closer to some mouse studies (that we cited, studies from the Dent lab). We have now cited Ojima and Horikawa (2016) and their earlier 2012 study in the revised manuscript (page 3, line 77; page 4, line 87; page 27, line 677).

Specific differences between our study and that of Ojima and Horikawa (2016) include: they used a single footstep sound was used as a target, and single exemplars of other natural sounds (clapping hands etc) were used non-targets. They used a competitive training paradigm. For the tempo shifts in particular, they observed a strong asymmetry between compression and expansion of sounds. Finally, they did not attempt to develop a model to explain GP behavior, which is the central point of our manuscript.

A simple frequency-based classifier can solve the basic categorization task, but does not match GP behavior, as described in the new analysis above.

Detailed commentsFigure 3 – what is the performance of the trained exemplars here? From the other figures I guess a lot higher than for the generalization stimuli; it looks like performance does take a hit on novel exemplars (possibly for GPs and the model?). The authors should justify why they didn't present stimuli as rare (and randomly rewarded) probe stimuli to assess innate rather than learned generalization.

On the last day of the training phase, the GPs achieved an average d′ of 1.94 and 1.9 for the chuts vs. purrs and wheeks vs. whines tasks respectively (now stated in page 6, line 148). This is somewhat greater than the d′ values for generalization (1.72 and 1.51 respectively). The d′ values do continue to improve slowly over the course of the other testing (for the unmanipulated stimuli in each set).

We initially presented novel stimuli as a separate block in order to minimize the number of trials required. We required about 200 trials to assess performance, and presenting these trials 20% of the time would have resulted in us having to present 1000 trials (5 days). But after receiving feedback at a conference, we indeed also tested generalization in the wheeks vs. whines task by presenting novel stimuli rarely (this was described in page 9, line 228 of the original manuscript, sentence beginning “*As an additional control…*”). We verified that GPs also generalized with this mode of stimulus presentation.

Figure 5. – segment length analysis – here the GP is much better at short segments. The model and the GP show a clear modulation in performance with segment length that is not picked up statistically due to an underpowered analysis. These data should either be analysed as a within animal logistic regression or an across animal mixed effect model. This could combine all 6 animals, with the task as a factor.

We thank the reviewer for this suggestion – we agree that the statistical analysis was likely under-powered. In the revised manuscript, we now use a generalized linear model with a logit link function, with animal ID as a random effect, to model both model and GP responses on a trial-by-trial basis. To evaluate the significance of the segment-length (and other manipulations), we compared this ‘full’ model to a ‘null’ model that only had stimulus type as a predictor and animal ID as a random effect. The new analysis does reveal, as expected by the reviewer, a statistically significant effect of segment length on both GP and model performance. These analyses are now described in page 13, line 315 and page 14, line 326 of the revised manuscript. Detailed methods are described in page 36, line 942.

Figure 6 – again, repeated measures anova on 7 conditions in 3 animals (χ2); the fact that the p-value is >0.05 means nothing! The real question – is the model quantitatively similar (could be addressed by assessing the likelihood of getting the same performance by generating a distribution of model performance on many model instantiations and then asking if the GP data falls within 95% of the observed values) or is it qualitatively similar? – for example show the same variation in performance across different call rates. In F the GP performance is robust across all durations, with a gradual increase in performance from the shortest call durations. In contrast the model peaks around the trained duration. In E both the model and the GP peak at just less than 1x call duration, but the model does really poorly at short durations. To me, this isn't 'qualitatively similar'.

Similar to the above model, we implemented a GLM to model both behavioral and model results on a trial-by-trial basis (page 15, line 352). This analysis revealed a weak effect of tempo shift on the behavioral performance of GPs for both the chuts vs. purrs and wheeks vs. whines tasks. Trial-by-trial fits to the model revealed a weak effect of tempo shift for chuts vs. purrs but no effect for wheeks vs. whines (page 16, line 370). In the revision, we state that there is a broad correspondence between model and data (page 16, line 379).

We performed similar GLM analyses for the SNR data (page 11, line 261) and F0 shift data (page 18, line 448) as well.

Figure 7 – why use the term chimeric in this figure and nowhere else? Are they not just the random ISI as in the second half of the figure?

Random ISI refers to replacing the ISIs of a call, say a purr, with ISIs drawn from the overall distribution of purr ISIs, i.e., the same call type. Chimeric calls are when ISIs of a purr are replaced with ISIs randomly drawn from the chut ISI distribution. The difference between random ISI and chimeric ISI is described in Figure 7’s legend as well as in the main text (page 16, 387 and page 16, line 395 respectively).

Figure 8 – I'm surprised at how poorly the model does on the reversed chuts. Again, the lack of a significant difference here cannot be used as evidence for equivalent performance.

We removed the phrase “Similar to GP behavior…” from our description of this result, and now simply state that “The model also maintained robust performance (*d′* > 1) for call reversal conditions but with an ~18% decrease in *d′* compared to behavior.” (page 18, line 426).

Figure 12 – I think I possibly don't understand this figure – how can the MIFs well suited for a purr vs chut distinction be different from a chut vs purr distinction? Either this needs elaboration or I think It would make far more sense to display chuts and whines (currently supplemental figure 12). Moreover, rather than these rather unintuitive bubble plots, a ranked list of each feature's contribution to the model, partial dependence plots, and individual conditional expectation plots would answer the question of which model features most contribute towards a guinea pig call categorization.

Each call type has associated with it ~20 MIFs that are best suited for distinguishing that call type from all other call types. When a stimulus is presented, we used template matching to detect the MIFs of all call types in the stimulus and weight the detected features by their log-likelihood ratios to provide the total evidence for the presence of a call type (Figure 2). Thus, when stimuli are presented (and their category known), we can determine the number of times a given MIF will be detected in within-category stimuli and ouside-category stimuli. The relative detection rate can be calculated from these values.

The reviewer is correct in pointing out that in the original manuscript, the main text describing Figure 12 used the Go/No-go terminology confusingly. We intended to mean “within-category” and “outside-category” stimuli. We have know rewritten the main text in the revised manuscript, and hope that this revised description is clearer (page 25, lines 589, 593).

There are 100 data points per panel in this figure, and a ranked list is also not easy to interpret. We realized that we had not uploaded the raw data underlying this figure as a supplement to the original manuscript. We have now uploaded an.xls file (Figure 12 – source data 1) containing a list of all MIF properties used in Figure 12, which can be sorted along multiple parameters as necessary. For visualization purposes, we have chosen to retain bubble plots from the original manuscript.

The model classifies the result based on weighted factors such as spectrotemporal information but does not consider specific biological neuronal networks that underly this categorization process. Moreover, the authors assert that intermediate-complexity features were responsible for call categorization; this assertion is expected as all MIFs selected are based on the inputted spectrotemporal data. In other words, the features in the classifier model could arguably said to be mid-complex as all vocalizations could be said to have inherent complexity. Together, the conclusions that the authors draw that this model could represent a biological basis for call categorization seem really to reach beyond the data; the model fits some aspects of the training set well but does not causally prove that GPs learn specific frequencies to represent calls early in their lives.

The reviewer is correct that we do not extensively comment on the biological basis of detecting MIFs or the complexity of the features. The reason for this is that this manuscript is the third in a series of studies that our lab has published on using such informative features for categorization. In the first study (Liu ST et al., Nat. Comm., 2019), we showed that restrictions to feature duration or bandwidth typically degraded model performance (Figure 7). We also showed that highly informative features tended to have intermediate complexity (Supplementary Figure 2). The second study more directly addressed the biological bases (Montes-Lourido et al., PLoS Biology, 2021), and showed that neurons in superficial cortical layers displayed receptive fields and call selectivity consistent with their being selective for some call features. Both these publications contain extensive discussions on the possible biological implementation of feature selectivity, and we did not want to repeat those points in the present manuscript. Instead, we briefly stated these results in the introduction section. In the revised manuscript, we have added a sentence describing the complexity results from the earlier study as well (page 2, line 49).

We agree with the reviewer that the model does not establish causality, nor do we claim in the manuscript that we are providing a causal explanation. As we state at the end of the Results section of the original (and revised) manuscript:

“Note that our contention is not that the precise MIFs obtained in our model are also the precise MIFs used by the GPs – indeed, we were able to train several distinct MIF sets that were equally proficient at categorizing calls. Rather, we are proposing a framework in which GPs learn intermediate-complexity features that account for within-category variability and best contrast a call category from all other categories, and similar to the model, recruit different subsets of these features to solve different categorization tasks.” (page 25, line 608)

Reviewer #3 (Recommendations for the authors):This is a great study! I appreciate the approach to complementing behavioral experiments with computational modelling.

Thank you for these kind words!

Considering the temporal stretching/compressing and the pitch change experiment: it remains unclear if the MIF kernels used by the models were just stretched/compressed to compensate for the changed auditory input. If so, the modelling results are kind of trivial. In this case, the model provides no mechanistic explanation of the underlying neural processes. This issue should at least be addressed in the corresponding Results section and might be an interesting cornerstone for a follow-up study.

As stated above, we did not alter the MIFs in any way for the tests – the MIFs were purely derived by training the animal on natural calls. In learning to generalize over the variability in natural calls, the model also achieved the ability to generalize over some manipulated stimuli. The fact that the model tracks GP behavior is a key observation supporting our argument that GPs also learn MIF-like features to accomplish call categorization.

We had mentioned at a few places that the model was only trained on natural calls. To add clarity, we have now included sentences in the time-compression and frequency-shifting results affirming that we did not manipulate the MIFs to match test stimuli. We also include a couple of sentences in the Discussion section’s first paragraph stating the above argument (page 26, line 627).